# Glucocorticoid receptor triggers a reversible drug-tolerant dormancy state with acquired therapeutic vulnerabilities in lung cancer

Stefan Prekovic [1,16✉], Karianne Schuurman[1,16], Isabel Mayayo-Peralta[1], Anna G. Manjón[2], Mark Buijs[1], Selçuk Yavuz [1], Max D. Wellenstein[3], Alejandro Barrera [4], Kim Monkhorst[5], Anne Huber[1,15], Ben Morris[6], Cor Lieftink[6], Theofilos Chalkiadakis[1], Ferhat Alkan [1], Joana Silva[1], Balázs Győrffy[7,8], Liesbeth Hoekman[9], Bram van den Broek [10], Hans Teunissen[11], Donna O. Debets[12], Tesa Severson[1], Jos Jonkers [13], Timothy Reddy [4], Karin E. de Visser [3], William Faller [1], Roderick Beijersbergen [6], Maarten Altelaar[9,12], Elzo de Wit [11], Rene Medema[2] & Wilbert Zwart [1,14✉]

The glucocorticoid receptor (GR) regulates gene expression, governing aspects of homeostasis, but is also involved in cancer. Pharmacological GR activation is frequently used to alleviate therapy-related side-effects. While prior studies have shown GR activation might also have anti-proliferative action on tumours, the underpinnings of glucocorticoid action and its direct effectors in non-lymphoid solid cancers remain elusive. Here, we study the mechanisms of glucocorticoid response, focusing on lung cancer. We show that GR activation induces reversible cancer cell dormancy characterised by anticancer drug tolerance, and activation of growth factor survival signalling accompanied by vulnerability to inhibitors. GR-induced dormancy is dependent on a single GR-target gene, *CDKN1C*, regulated through chromatin looping of a GR-occupied upstream distal enhancer in a SWI/SNF-dependent fashion. These insights illustrate the importance of GR signalling in non-lymphoid solid cancer biology, particularly in lung cancer, and warrant caution for use of glucocorticoids in treatment of anticancer therapy related side-effects.

---

[1] Division of Oncogenomics, Oncode Institute, The Netherlands Cancer Institute, Amsterdam, The Netherlands. [2] Division of Cell Biology, Oncode Institute, The Netherlands Cancer Institute, Amsterdam, The Netherlands. [3] Division of Tumour Biology and Immunology, Oncode Institute, The Netherlands Cancer Institute, Amsterdam, The Netherlands. [4] Department of Biostatistics & Bioinformatics, and Centre for Genomic & Computational Biology, Duke University Medical Centre, Durham, NC, USA. [5] Department of Pathology, The Netherlands Cancer Institute, Amsterdam, The Netherlands. [6] Division of Molecular Carcinogenesis and Robotics and Screening Centre, Netherlands Cancer Institute, Amsterdam, The Netherlands. [7] Semmelweis University Department of Bioinformatics and 2nd Department of Pediatrics, Budapest, Hungary. [8] TTK Cancer Biomarker Research Group, Institute of Enzymology, Budapest, Hungary. [9] Mass spectrometry/Proteomics Facility, The Netherlands Cancer Institute, Amsterdam, The Netherlands. [10] Division of Cell Biology and BioImaging Facility, The Netherlands Cancer Institute, Amsterdam, The Netherlands. [11] Division of Gene Regulation, Oncode Institute, The Netherlands Cancer Institute, Amsterdam, The Netherlands. [12] Biomolecular Mass Spectrometry and Proteomics, Bijvoet Center for Biomolecular Research and Utrecht Institute for Pharmaceutical Sciences, Utrecht University, Utrecht, The Netherlands. [13] Division of Molecular Pathology, Oncode Institute, The Netherlands Cancer Institute, Amsterdam, The Netherlands. [14] Laboratory of Chemical Biology and Institute for Complex Molecular Systems, Department of Biomedical Engineering, Eindhoven University of Technology, Eindhoven, The Netherlands. [15] Present address: Olivia Newton-John Cancer Research Institute and School of Cancer Medicine, La Trobe University, Melbourne, VIC, Australia. [16] These authors contributed equally: Stefan Prekovic, Karianne Schuurman. ✉email: s.prekovic@nki.nl; w.zwart@nki.nl

The glucocorticoid receptor (GR) is a member of the nuclear hormone receptor superfamily and a ligand-activated transcription factor[1]. This multidomain protein exerts its function through chromatin binding and communication with the transcription machinery, ultimately modulating the expression of a large number of genes, across diverse cell types[2,3]. As a homeostatic regulator, GR has an imperative role in neuroendocrine integration, circadian rhythm, immune system control and glucose metabolism[4]. The action of this transcription factor extends beyond general physiology as its impact can be seen in various disease types, including cancer[5].

While pharmacological agonists of the GR (e.g., prednisone and dexamethasone) have been intensively used as therapeutics in the treatment of lymphoid cancers, for non-lymphoid solid (i.e., non-haematologic) cancer patients they are utilised solely as an adjuvant treatment to alleviate symptoms caused by anticancer therapy. However, studies on in vitro and in vivo models of numerous non-lymphoid solid cancer types (e.g., prostate, lung and breast cancer) have shown that glucocorticoids (GCs) decrease cancer incidence and reduce the growth of cancer[6–13]. In addition, in aged mouse haploinsufficiency models, GR loss predisposes tumour development across multiple organ systems[14]. Despite these observations, a precise mode-of-action through which GCs affect non-lymphoid solid cancers remains unclear.

Herein, we elucidate the molecular mechanisms by which GR activation blocks cell proliferation in non-lymphoid solid cancers with the primary focus on lung cancer. We demonstrate that GR activation induces cancer cell dormancy, accompanied by a diminished response to a large array of anticancer drugs, activation of growth factor survival signalling (IGF-1R) and acquisition of vulnerability to IGF-1R inhibitors in cell lines and xenograft models. Furthermore, we reveal that this phenotype is dependent on GR-mediated regulation of CDKN1C (which encodes for p57) in a SWI/SNF-dependent fashion through long-range genomic regulation of an upstream distal enhancer. Ultimately, using transcriptomics and chromatin accessibility data of clinical samples, we show that this mode of regulation occurs in multiple human non-lymphoid solid cancer types.

## Results

**Stress hormone receptor activation leads to cell dormancy**. In order to study the phenotypic and genotypic consequences of GR activation, five non-small cell lung cancer models (Supplementary Fig. 1a) were selected based on their steroid hormone receptor expression profiles[15,16]. Expression of GR was confirmed by western blot analysis (Fig. 1a), demonstrating comparable expression levels across five cell lines. The GC treatment (specific treatment information per experiment can be found in Supplementary Table 1) of A549, H2122 and H1944 led to a significant reduction in proliferation rate as observed in live-cell tracking experiments using SiR-DNA (Fig. 1b). Conversely, growth rates of H1975 and H460 cell lines were unaffected by GC therapy (Fig. 1b). The propidium iodide-staining and subsequent flow cytometry analysis revealed that the drop in proliferation rate was underlined by a reduction in the S phase and an increase in the G0/G1 phase of the cell cycle (Supplementary Fig. 1b). Treatment with GCs did not induce apoptosis, as demonstrated by the absence of cleaved PARP detected by means of western blot analysis (Supplementary Fig. 1c).

In agreement with the observed growth arrest upon GC treatment, a high degree of protein dephosphorylation was observed (Fig. 1c), most of which were involved in direct regulation of transcription and cell cycle as evidenced by gene-set analysis (Fig. 1d). This was accompanied by a strong, significant

downregulation of E2F targets (Hallmark gene sets; M5925) on the whole-proteome level (Fig. 1e). Conversely, the phospho-proteomes of the H1975 and H460, cell lines that are not growth-arrested by GCs, were not significantly altered by GC treatment (Supplementary Fig. 1d).

As lack of cleaved PARP suggested that growth arrest does not involve apoptosis (Supplementary Fig. 1c), we inspected whether GC treatment led to the acquisition of senescence. Firstly, we observed a significant (FDR $q$ value: A549 = 0.007; H2122 = 0.019; H1944 = 0.05) enrichment score for senescence gene-set (Fridman Senescence Signature, M9143) on whole-proteome level (Fig. 1f) in GC-treated cells compared to the vehicle-treated conditions. In agreement with this, we detected positive staining for senescence-associated β-galactosidase upon GC stimulation in A549, H2122 and H1944, but not H1975 and H460 (Fig. 1g, h). However, cell cycle exit was neither accompanied by changes in p53 protein expression levels (Supplementary Fig. 1c) nor activation of the p53 pathway as shown by gene-set enrichment analysis of RNA sequencing and full proteome datasets (Supplementary Fig. 1e). Upon ligand withdrawal, the growth inhibition was lost and cells restarted proliferating (Supplementary Fig. 1f). In addition, a decrease was observed in the overall metabolic activity/capacity, as evidenced by a significant decrease in oxygen consumption rate (reflecting mitochondrial respiration; Fig. 1i and S1g), and extracellular acidification rate (reflecting glycolytic output; Fig. 1j).

Furthermore, we investigated the gene signatures of cell cycle and senescence in human lung adenocarcinoma tumours stratified on the basis of GR activity (calculated as $Z$ score of 253 genes associated with GR activation; 25% split). In support of our experimental findings, The Cancer Genome Atlas (TCGA)-based analysis revealed that human lung tumours with high GR activity have higher expression level of senescence-associated genes, and lower expression of cell cycle-related genes in comparison to tumours with low GR activity (Supplementary Fig. 1h, i). Importantly, Kaplan–Meier survival analysis was performed on data from 1529 lung cancer patients, of whom the majority did not receive (neo)adjuvant therapy before and after surgery[17–22]. The patients were divided into three groups based on transcriptomics-derived GR activity and the Kaplan–Meier analysis demonstrated that patients with high GR activity have a more favourable outcome based on overall survival (Supplementary Fig. 1j, l) and recurrence-free survival (Supplementary Fig. 1k, m) probabilities than patients with intermediate or low levels of GR activity.

Taken together, we conclude that GCs induce a transition to a dormant, reversible cellular state. Importantly, the induction of growth arrest by stress hormone receptor activation extends to other non-lymphoid solid cancer types, as this was also observed in primary patient-derived and pre-established models of mesothelioma; a cancer type derived from cells of mesodermal lineage (Supplementary Fig. 2).

**Glucocorticoid-induced cell dormancy is characterised by anticancer drug tolerance and activation of IGF-1R survival signalling**. To further characterise the GC-induced cell dormancy and the underlying molecular pathways that support survival, we performed a drug screen (2277 compounds from diverse sub-libraries) in the H1944 cell line. The cells were cultured in the presence or absence of GCs for 2 days, then divided over different arms of the screen (Supplementary Fig. 3a)—(1) vehicle arm, (2) GC pre-treated arm in which GC treatment was added before and continued throughout the screen and (3) a GC-co-treatment arm in which GCs were added at the same time as the library compounds. For all arms, the library compounds were used at two

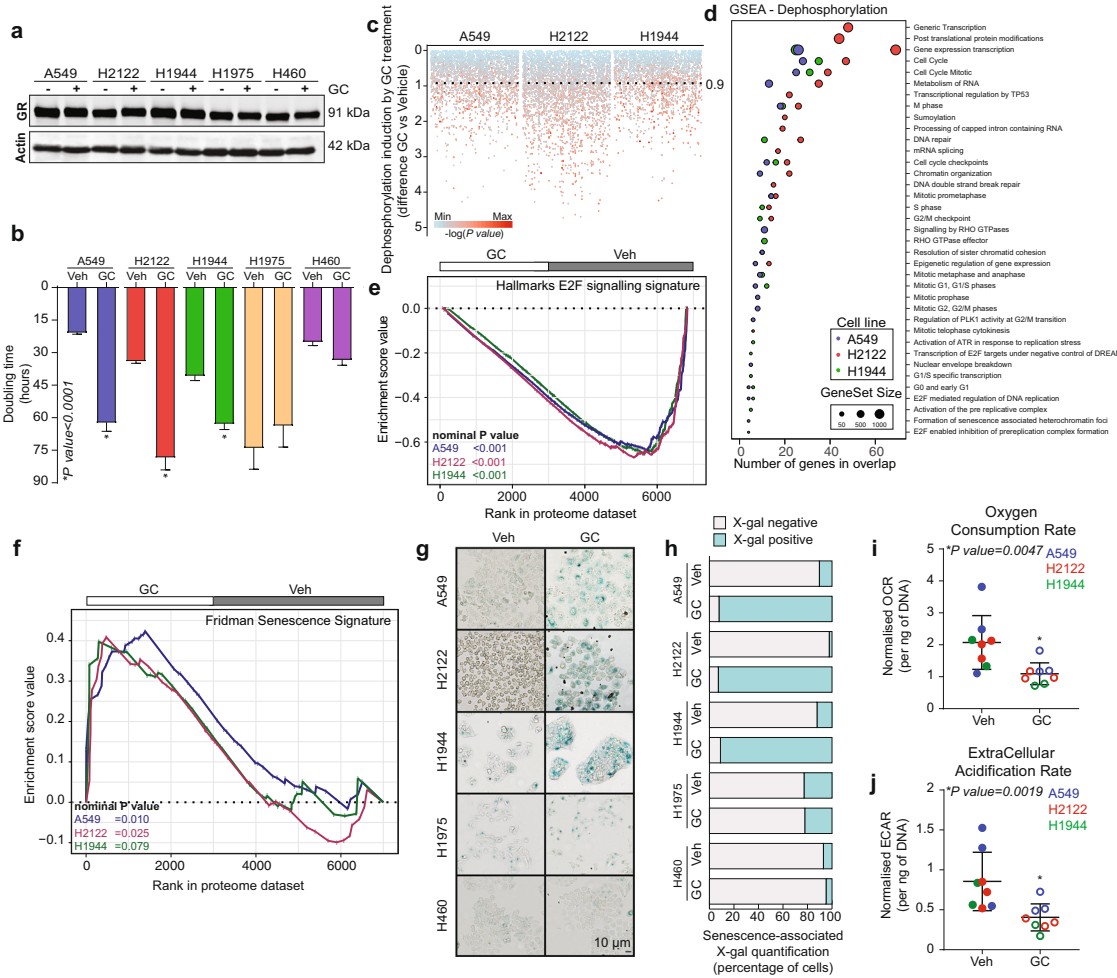

**Fig. 1 Glucocorticoid receptor induces cell dormancy in lung cancer models. a** Representative western blot showing expression of GR, with actin as a loading control ($n = 2$). **b** Doubling time (in hours) as calculated from real-time proliferation data of cells treated with vehicle (Veh) or glucocorticoids (GC). Bars depict mean value ± SEM of independent doubling time calculations (A549 $n = 18$, H2122 $n = 18$, H1944 $n = 18$, H1975 $n = 24$, H460 $n = 18$). $P$ values were determined by two-sided Mann–Whitney test. **c** All GC-induced dephosphorylation events captured by mass spectrometry across A549, H2122 and H1944, relative to untreated condition ($n = 3$). $P$ values were determined by two-sided $t$ test. **d** Gene-set-enrichment analysis (GSEA) of dephosphorylated events discovered by mass spectrometry analysis performed in A549 (blue), H2122 (red) and H1944 (green) cell lines ($n = 3$). **e** E2F target gene-set (M5925) GSEA enrichment profiles for whole-proteome mass spectrometry experiments performed in A549 (blue), H2122 (red) and H1944 (green) cell lines ($n = 3$). Nominal $P$ values were determined by GSEA software. **f** Fridman senescence gene-set (M9143) GSEA enrichment profiles for whole-proteome mass spectrometry experiments performed in A549 (blue), H2122 (red), and H1944 (green) cell lines ($n = 3$). Nominal $P$ values were determined by GSEA software. **g** Representative images of senescence-associated β-galactosidase (X-gal) stained cells, untreated (Veh) or glucocorticoid-treated (GC) ($n = 3$). Scale bar, 10 µm. **h** Quantification of at least 200 cells from senescence-associated β-galactosidase experiments represented in fraction of negative (grey) and positive cells (blue) ($n = 3$). **i** Per DNA content normalised oxygen consumption rate (OCR) for A549, H2122 and H1944, without (Veh) or with glucocorticoids (GC). Bars depict mean values ± SEM of independent experiments (A549 $n = 3$, H2122 $n = 3$, H1944 $n = 2$). $P$ values were determined by two-sided Mann–Whitney $U$ test. **j** Per DNA content normalised extracellular acidification rate (ECAR) for A549, H2122 and H1944, without (Veh) or with glucocorticoids (GC). Bars depict mean values ± SEM of independent experiments (A549 $n = 3$, H2122 $n = 3$, H1944 $n = 2$). $P$ values were determined by two-sided Mann–Whitney $U$ test. Source data are provided as a Source Data file.

different concentrations—1 µM and 5 µM. After 6 days of exposure to the library drugs, cell viability was assessed using a CellTiter-Blue assay (Supplementary Fig. 3a), and GC arms were compared to the vehicle arm.

Firstly, GCs decreased the sensitivity to numerous drugs in both GC pre-treatment and co-treatment arms (Fig. 2a and Supplementary Data 1). Using a selected array of drugs based on the first screen, we have found that GCs significantly reduced the effectiveness of these drugs in another lung cancer model, the H2122 cell line (Fig. 2b). By means of Compound Set Enrichment Analysis within the CSgator, a comprehensive analytic tool for setwise interpretation of compounds[23], we reveal that compounds with reduced effectiveness after/during GC treatment were

predominantly threonine protease, kinase, guanylate cyclase and structural protein inhibitors (Fig. 2c). Importantly, we show that among these are various drugs clinically approved for the treatment of lung cancer, including vinorelbine tartrate, dabrafenib, trametinib and docetaxel (Supplementary Fig. 3b). Secondly, the GR activation also increased sensitivity to nine inhibitors. More specifically, all identified compounds with a significant degree of drug response enhancement ($P$adj $< 0.05$ and difference $< -0.3$) on GR-treatment were classified as IGF pathway inhibitors (Fig. 2a and Supplementary Fig. 3c). We have successfully validated these findings across the three cell lines (A549, H2122 and H1944) using a logarithmic range of concentrations and drug stock obtained from a different supplier

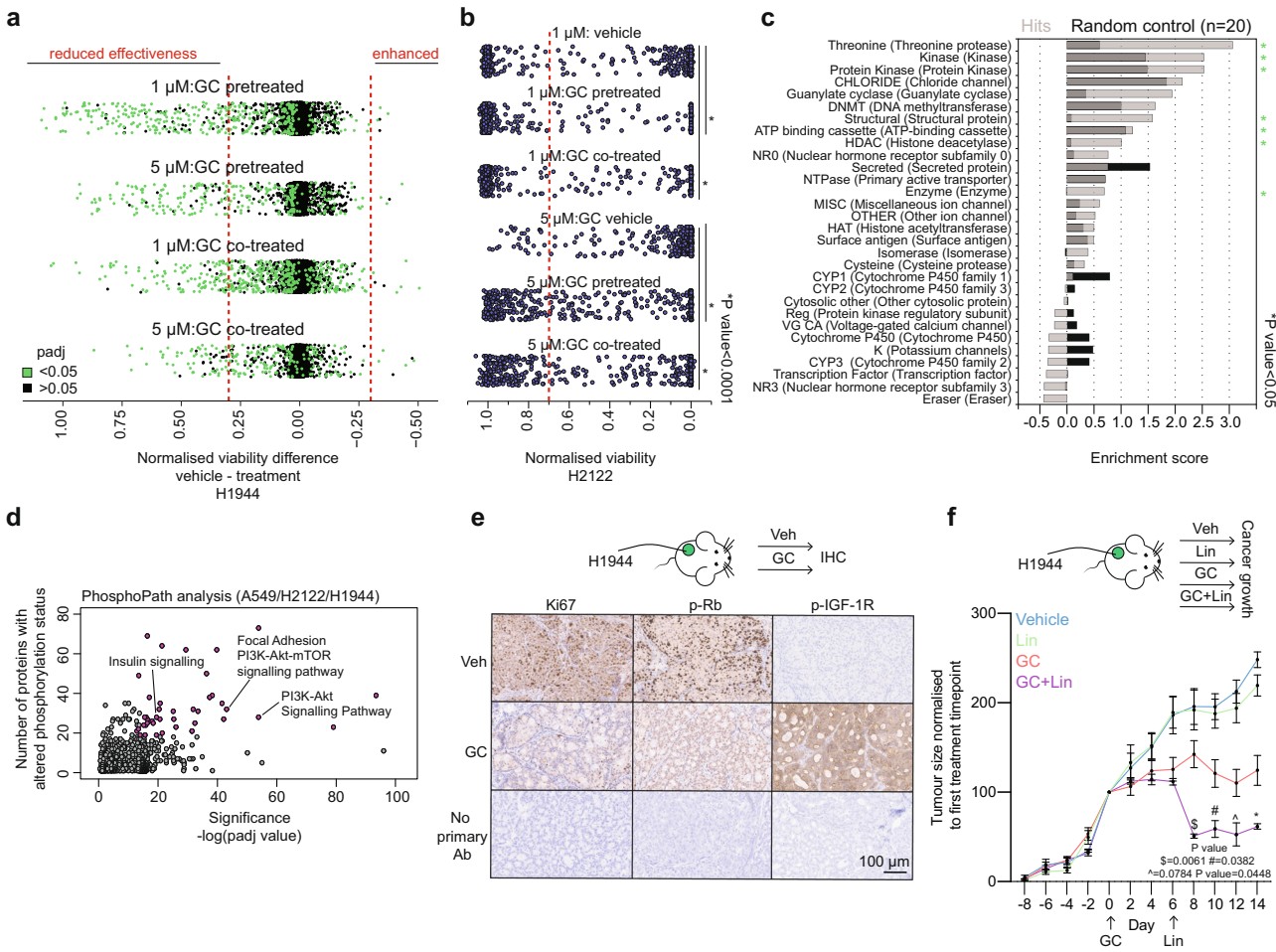

**Fig. 2 Cell dormancy phenotype is accompanied by drug tolerance and viability maintenance via growth factor signalling activation. a** Scatter plot showing viability differences of vehicle and glucocorticoid pre-treated or co-treated screen arms with either 1 μM and 5 μM of the screening drugs. Compounds having significant differential viability are depicted in green (Padj < 0.05). Adjusted P values (Padj) were determined by two-sided t test with multiple testing correction (Benjamini–Hochberg method). **b** Normalised viability of H2122 cell lines in response to compounds found to diminish viability (<0.7) of vehicle arm of H1944 cell lines in the first screen at 5 μM concentration. P values were determined by two-sided Welch's t test. **c** Compound Set Enrichment Analysis data computed using CSgator software of drugs with reduced efficacy upon GC treatment. Random controls of the same size were generated to compute background enrichment (n = 20). P values (FDR-adjusted) were determined by CSgator software. **d** PhosphoPath pathway analysis of phosphoproteomics data depicting signalling changes in glucocorticoid-treated cells (A549, H2122 and H1944) (n = 3). P values were determined by PhosphoPath software. **e** Representative images showing immunohistochemical stainings of xenograft cancer samples obtained after treatment of NOD-SCID-γ mice with either vehicle (Veh) or glucocorticoids (GC) (n = 4). Primary antibody was omitted as staining control. Scale bar, 100 μm. **f** Normalised tumour growth in xenograft models of H1944 cells in NOD-SCID-γ mice treated with either vehicle (Veh = blue), linsitinib (Lin = green), glucocorticoids (GC = red), or combination (GC + Lin = purple). Arrows indicate when treatment was started. Mean values ± SEM depicted (Dexa + Lin n = 4, Dexa n = 6, Veh n = 5, Lin n = 6 animals). P values were determined by mixed-model ANOVA (Tukey's multiple comparison test).

(Supplementary Fig. 3d). The modest response to GC/IGF-1R inhibitor combination in H2122 may be explained by the pre-existing dependency of this cell line on IGF-1R signalling in absence of GCs, as suggested by the data available on The Cancer Dependency Map portal[24] (Supplementary Fig. 3e). In conjunction with induced vulnerability to IGF-1R inhibitors, phospho-pathway analysis of the three cell lines revealed significant changes in the protein phosphorylation status of insulin signalling (including IGF-1R protein) and the related downstream pathways[25,26], confirming the implied increase in their activity upon GR activation (Fig. 2d).

Treatment of xenograft animals with GCs elicited a pronounced effect on the transcriptome of the engrafted H1944 tumours (Supplementary Fig. 3f). Activation of GR increased gene expression of its target genes, including *GILZ1*, *FKBP5* and *PER1* (Supplementary Fig. 3g). Interestingly, we observed that metastasis-associated genes *MYCN*, *ID4* and *VCAM1* were

significantly downregulated by GC treatment (Supplementary Fig. 3g). Furthermore, we confirmed our previous findings from in vitro experiments, showing that GR activation leads to a significant enrichment of the GR activity signature and Fridman Senescence signature, and significant downregulation of genes involved in E2F signalling (Supplementary Fig. 3h). In agreement with this, the treatment of animals bearing H1944 xenograft tumours with GCs led to a decrease in Ki67 immunostaining and retinoblastoma (Rb) phosphorylation (Fig. 2e; no changes in total Rb levels (Supplementary Fig. 3i)), without induction of p21 expression and apoptosis (Supplementary Fig. 3i), as seen by the absence of cleaved Caspase-3 signal. Importantly, activation of GR promoted phosphorylation of IGF-1R in xenograft tumours (Fig. 2e). In line with this, GC-induced IGF-1R inhibitor vulnerability was tested in vivo, in NOD-SCID-γ xenograft models of the H1944 cell line. Treatment with GCs led to stable tumour growth arrest in comparison to the vehicle-treated mice

(Fig. 2f). The addition of IGF-1R inhibitor linisitinb to treatment schedule of these mice led to a sharp and sustained decrease in tumour size (Fig. 2f). We successfully validated these findings in an A549 cells xenograft model in NOD-SCID-γ mice (Supplementary Fig. 4a, b). In animals bearing A549 xenografts, we demonstrated that the combination of two different IGF-1R inhibitors (linsitinib and GSK1838705A) with GCs had a significant effect on tumour size in comparison to the GC-monotherapy arm (Supplementary Fig. 4a, b).

To investigate if GR directly contributes to the modulation of IGF-1R signalling, we explored the GC-treatment time-course RNA sequencing dataset in A549 cell line[27]. In the insulin signalling gene-set (M18155), exclusively FOXO1, IRS2 and PYGB (glycogen phosphorylase; not part of canonical IGF-1R signalling[28]) were stably upregulated by GCs (Supplementary Fig. 4c). We focused on inspecting the GR regulation of two genes directly involved in the IGF-1R pathway—FOXO1[29] and IRS2[30]. For this, we made use of GR ChIP sequencing and Hi-C time-course data from A549 cells[27]. GR chromatin binding to several enhancer sites within the corresponding topologically associating domains (TADs) (Supplementary Fig. 4d, e) containing the FOXO1 and IRS2 gene loci was observed (Supplementary Fig. 4f, g, left). In addition to binding of GR to these enhancers, induction of a single enhancer–promoter loop containing two GR-binding sites in the loop anchor for the FOXO1 gene was observed (Supplementary Fig. 4f, right). As for IRS2, a complex web of six enhancer–promoter loops was detected, containing nine GR-binding sites (Supplementary Fig. 4g, right).

Collectively, these data show that activation of GR with GCs induces broad tolerance to anticancer drugs and that viability of GC-induced dormant cells is maintained via engagement of the IGF-1R signalling pathway.

**Cell cycle inhibitor p57 is necessary for glucocorticoid-induced cell dormancy**. In order to identify the driver of GC-induced cell dormancy, we performed RNA sequencing in the A549, H2122 and H1944 cell lines treated with vehicle or GCs for 8 h. Comparison of GC-induced transcriptional modulation across the cell lines revealed a high degree of similarity (Fig. 3a). A focused analysis on genes differentially expressed upon GR activation ($-2 \leq \log2$ fold $\geq 2$ and $P$adj $\leq 0.01$) revealed 65 genes shared between the three cell lines (Fig. 3b); with only one being a cell cycle regulator; CDKN1C (which encodes for p57). In addition, this gene was found upregulated in the H2795 mesothelioma cell line which was growth-arrested by GCs, but not in two GC-resistant mesothelioma models (Supplementary Fig. 5a).

Expression of p57 was analysed by immunofluorescence and western blot. GC-dependent induction of p57 and its nuclear localisation were found exclusively in the dormant condition (A549, H2122 and H1944), while not detected in the GC-unresponsive H1975 and H460 models (Fig. 3c and S5b). Furthermore, in line with its well-described cell cycle inhibitory function[31] rapid immunoprecipitation of endogenous proteins (RIME)[32] in H2122 cells demonstrated that p57 interacts with various CDKs (CDK 1, 2, 4 and 6) as well as other cell cycle-related proteins such as CCNB1, not previously reported as p57 interacting protein (Fig. 3d and Supplementary Data 2). Importantly, upregulation of CDKN1C mRNA (Fig. 3e) preceded the transcriptional downregulation of various cell cycle genes (including CCND3 and CCNE2) which was observed after 4 h of GC treatment (Fig. 3f). Cumulatively, these data suggest that p57 may be involved in initiation of dormancy upon GR activation.

To address whether p57 is required for GR-induced cell cycle exit, we performed CRISPR-Cas9-mediated disruption of the CDKN1C gene in the A549, H2122 and H1944 cell lines. While GR nuclear translocation following GC treatment was not affected, induction of p57 expression in a polyclonal CDKN1C knockout (p57-KO) population was greatly diminished (Fig. 3g and Supplementary Fig. 5c). To inspect whether GCs are still able to induce cell dormancy in p57-KO cells, live-cell imaging of SiR-DNA-stained cells with and without GCs was performed, and the number of cells undergoing mitosis in the first 60 h of treatment quantified. In agreement with our hypothesis, the genetic disruption of the CDKN1C gene was sufficient to diminish cell dormancy induction by GCs (Fig. 3h). In addition, senescence-associated β-galactosidase staining was performed after treatment with vehicle or GCs in H2122 p57-WT and p57-KO. The senescence-associated staining was strongly decreased in p57 knockout cells (Supplementary Fig. 5d), confirming the critical role of p57 in dormancy induction. To further test if GR-mediated upregulation of CDKN1C drives transcriptional changes leading to growth arrest, we investigated the transcriptomic differences of the p57-WT and p57-KO H2122 cells. In the absence of GCs, no statistically significant differences in mRNA expression were detected between the p57-WT and p57-KO H2122 cells (Supplementary Fig. 5e, f). While CRISPR-Cas9-mediated disruption of p57 did not alter transcriptional modulation of active GR-associated genes (Fig. 3i and Supplementary Fig. 5g), it diminished downregulation of genes involved in cell cycle (Supplementary Fig. 5g–i). In conjunction with this, the changes in gene-set enrichment analysis of E2F targets (Fig. 3j) and cell cycle-related genes (Fig. 3k) typically induced by GCs were not observed in the p57-KO model, confirming the hypothesis that p57 upregulation is necessary for the growth-arrest phenotype.

Taken together, our data show that direct GR-mediated upregulation of a single gene (CDKN1C) is required to initiate growth arrest in human lung cancer cell line models.

**Glucocorticoid receptor regulates CDKN1C expression through a previously uncharacterised distal enhancer**. The regulation of CDKN1C by enhancers has been under debate and the precise enhancers controlling its expression in human cells remain unknown[33,34]. To address if CDKN1C upregulation is directly dependent on GC-mediated activation of GR instead of an off-target effect of the ligand (e.g., activation of mineralocorticoid receptor), we generated GR knockout (GR-KO) H2122 cell lines. In H2122 GR-WT cells, nuclear localisation of GR and a concomitant expression of p57 was observed upon GC treatment, while in the polyclonal GR-KO cell population no signal for GR nor p57 was detected (Supplementary Fig. 6a).

Therefore, we sought to establish direct regulation of the CDKN1C gene by GR and to elaborate on the mechanism using ChIP sequencing. We observed chromatin binding of GR at three different sites (Enhancers 1, 2 and 3) located within the topologically associating domain (TAD) region containing a large part of KCNQ1 and the entire CDKN1C gene, flanked by CTCF sites as determined by Hi-C and ChIP-sequencing analysis (Fig. 4a). Interestingly, GR chromatin binding, as established by ChIP sequencing, was not detected at the CDKN1C promoter, contrasting a prior electrophoretic mobility shift assay (EMSA)-based study[35]. This discrepancy may potentially be explained by the absence of chromatin context in EMSA experiments. The active enhancer-associated factors[36], histone acetyltransferase p300 and H3K27Ac chromatin mark, were most pronounced at Enhancer 1 (Fig. 4b). In addition, cohesin (SMC3/Rad21) recruitment, known to be crucial for enhancer–promoter contacts, was observed at Enhancer 1 and the CDKN1C promoter (Fig. 4b). The intra-TAD localisation, GR binding, p300

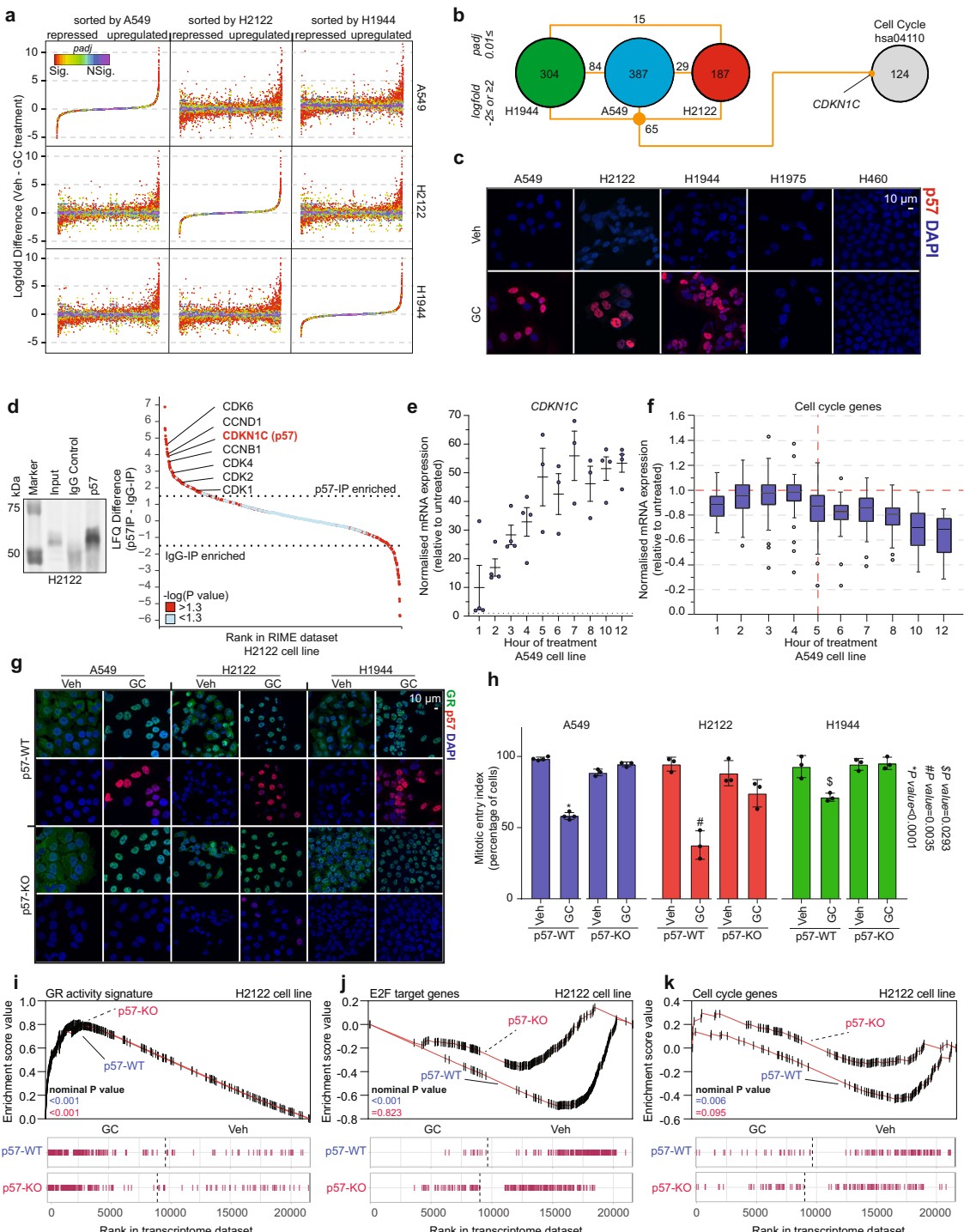

recruitment, strong H3K27Ac signal and cohesin localisation, all suggest that Enhancer 1 (hereinafter referred to as CERES (*CDKN1C* Enhancer Regulated by Steroids)) is the main regulatory element through which GR regulates *CDKN1C* gene expression, while Enhancers 2 and 3 could potentially serve as auxiliary enhancers.

To establish whether these particular enhancers and the *CDKN1C* locus are in proximity to one another in 3D genome space, we performed 4C-seq experiments[37]. The unbiased interaction analyses from the viewpoint of the *CDKN1C* promoter revealed that GC treatment enhanced the interaction with two distal regions within the *KCNQ1* gene (*a* and *b*) (Fig. 4c). Regions

*a* and *b* coincided with locations of CERES and enhancers 2/3, respectively. To unequivocally show that signal originating from the region *a* is driven by close proximity of *CDKN1C* promoter and CERES, we performed the reciprocal 4C-seq experiment from the CERES viewpoint. In A549, H2122 and H1944, we observed a statistically significant enhancement of the contact between this enhancer and *CDKN1C* promoter by GCs (Fig. 4d). Conversely, this enhancement was absent in two GC-unresponsive models of lung cancer, H1975 and H460 cell lines (Supplementary Fig. 6b).

To investigate the contribution of individual enhancers to the GR-induced *CDKN1C* gene upregulation, we performed CRISPR-Cas9 experiments to excise the individual enhancer elements

**Fig. 3 p57 expression is required for GC-induced growth arrest. a** Comparison of GC-induced transcriptional changes in RNA sequencing experiments across three cell lines (A549, H2122 and H1944) that are growth-arrested by glucocorticoids ($n = 2$). Adjusted $P$ values ($P$adj) were determined by DESeq2 (Wald test $P$ values corrected for multiple testing using Benjamini and Hochberg method). **b** Intersect of genes differentially expressed upon GC treatment with a cell cycle gene set (hsa04110). **c** Representative immunofluorescence images showing expression and localisation of p57 (red), with DAPI as nuclear staining (blue) ($n = 3$). Scale bar, 10 μm. **d** Western blot for p57-IP experiments in H2122 cell line treated with GCs (left). Waterfall plot depicting p57-IP enrichment over IgG control in H2122 cell line treated with GCs (right). Proteins considered to be interacting with p57 are 1.5 LFQ enriched over IgG (dotted line) and significant ($-\log(P \text{ value}) > 1.3$; red) ($n = 4$). $P$ values were determined by two-sided $t$ test. **e** Normalised *CDKN1C* mRNA expression level throughout the time-course experiment with glucocorticoids in A549 cells (ENCSR897XFT). Mean values ± SEM of independent biological replicates are depicted ($n = 4$; for timepoints 5, 6, 7 and 8, $n = 3$). **f** Normalised mRNA expression level of cell cycle genes (hsa04110; $n = 125$ genes) throughout the glucocorticoid-treatment time course in A549 cells (ENCSR897XFT). Box plot depicting expression of cell cycle genes based on at least three biological replicates per timepoint is depicted. The central mark indicates the median, and the bottom and top edges of the box indicate the 25th and 75th percentiles, respectively. The maximum whisker lengths are specified as 1.5 times the interquartile range and outliers are depicted as empty circles. **g** Representative immunofluorescence images showing expression and localisation of GR (green), p57 (red) in p57-WT and p57-KO cell lines. DAPI is used as nuclear staining (blue) ($n = 3$). Scale bar, 10 μm. **h** Percentage of cells undergoing mitosis in a 60 h real-time imaging experiment under vehicle (Veh) and glucocorticoid (GC) treatment. Mean values ± SD of independent biological replicates is depicted (A549 $n = 4$, H2122 $n = 3$, H1944 $n = 3$). $P$ values were determined by two-sided Welch's $t$ test. **i–k** GR activity signature, E2F target (M5925), and cell cycle (hsa04110) gene signature GSEA enrichment profiles for whole-transcriptome experiments of p57-WT and p57-KO H2122 cells, GC-treated (GC) compared to untreated (Veh) ($n = 2$). Nominal $P$ values were determined by GSEA software.

from the genome. Using pairs of guide RNAs, we excised either the CERES, E2 or E3 enhancer. In addition, we excised the *CDKN1C* gene and *ABCB1* promoter as positive and negative controls, respectively. Upon excision of the *CDKN1C* gene and the CERES enhancer in a polyclonal cell population, we observed a significant decrease in *CDKN1C* upregulation, (Fig. 4e) and rescue from growth arrest (Fig. 4f) in comparison to the negative control condition. This was not observed in the E2 and E3 deletion experiments, where induction of *CDKN1C* (Fig. 4e) and the degree of growth arrest (Fig. 4f) were comparable to the ones of the control cell lines. These experiments show that the CERES enhancer is required for a robust upregulation of *CDKN1C* by GR and therefore transition to a dormant state.

Collectively, we have discovered a GR-driven enhancer that regulates *CDKN1C* gene through long-distance chromatin interactions, thereby controlling cell dormancy entry.

**SWI/SNF complex is an integral part of a proficient GR transcriptional machinery controlling the expression of *CDKN1C*.** To gain more insight into the mechanism of GR regulation of *CDKN1C*, we compared cell lines in which GR was able to induce dormancy (A549, H2122 and H1944) to the ones in which it cannot (H1975 and H460). Across all the cell lines used, GR was able to readily translocate to the nucleus in response to GCs (Fig. 5a) and effectively bind thousands of sites in the genome (Fig. 5b), as demonstrated by immunofluorescence and ChIP-sequencing experiments, respectively. In contrast to that, GR-driven gene expression changes were observed in the GC-growth-arrested cell lines (A549, H2122 and H1944), while this was strongly attenuated in the GC-unresponsive cell lines H1975 and H460 (Fig. 5c). As co-regulator recruitment is imperative for transcriptional modulation, we subsequently investigated the molecular composition of the GR transcriptional complex by performing RIME[32]. In the dormancy-induced cell lines, GR was able to successfully recruit numerous proteins to its complex, including *NCOA1* and *NRIP1* (Fig. 5d and Supplementary Data 3), previously reported to be critical for GR-driven transcriptional regulation[38]. Despite the ability to bind chromatin (Fig. 5b), GR was unable to stably recruit coregulators in H1975 and H460 (Fig. 5d and Table S4). To unravel the composition of the active chromatin-bound GR complex, we performed a statistical comparison of GR-active (A549/H2122/H1944) and GR-inactive (H1975/H450) cell lines (Supplementary Data 4). While

GR itself was detected at comparable levels (Fig. 5e), pathway enrichment analysis (Gene Ontology gene-sets) revealed that an active-GR interactome is composed out of four major parts that include the nuclear transcription factor complex (nominal $P$ value = 0.018), SWI/SNF complex (nominal $P$ value < 0.001), mediator complex (nominal $P$ value = 0.056), and the RNA polymerase II complex (nominal $P$ value = 0.001) (Fig. 5f).

The SWI/SNF chromatin remodelling complex was of particular interest, as transcriptional downregulation of its members has been associated with GC-resistance in human acute lymphoblastic leukaemia[39,40]. Firstly, we interrogated whether GR activity is affected in human lung tumours bearing deleterious mutations in the members of the SWI/SNF complex. For this, we made use of a GR activity score (explained above) and the lung adenocarcinoma dataset from TCGA (91/877 tumours harboured SWI/SNF mutations; *SMARCB1* 18.09%; *SMARCC2* 9.52%, *SMARCD2* 6.66%, *SMARCD3* 1.90%, *ARID1A* 29.52%, *ARID2* 31.42%, *SMARCE1* 2.85%). The GR activity score was significantly lower in the tumours bearing SWI/SNF mutations (Fig. 5g), suggesting that these may influence GR activity. Using publicly available data of GC-growth-arrested cervical cancer cell line model (HeLa) that upregulate *CDKN1C* upon GC treatment (Supplementary Fig. 7a–c), we observed binding of GR and multiple SWI/SNF members to the CERES enhancer (Supplementary Fig. 7d), suggesting that this mechanism is active in other cancer types. To experimentally test if a causal relationship between SWI/SNF complex and GR activity exists, we performed short hairpin (shRNA) mediated knockdown (at least two shRNAs per target) of each SWI/SNF complex member in the H2122 cell line. Efficient knockdown for all eight SWI/SNF components was confirmed using RT-qPCR (Fig. 5h, left). Following this, we treated the knockdown models with GCs and performed RT-qPCR analysis for *CDKN1C* and housekeeping reference genes. Interestingly, while knockdown of *ARID1A*, *SMARCE1*, *SMARCA2* and *SMARCB1* had a negative impact on GR-mediated *CDKN1C* expression, loss of *SMARCC2* and *SMARCD2* further boosted of GR-induced upregulation of *CDKN1C* (Fig. 5h, middle). The knockdown of *ARID2* and *SMARCD3* had no impact on *CDKN1C* upregulation (Fig. 5h, middle). To confirm these findings on protein level, we performed an immunofluorescence staining of p57 in GC-treated condition and quantified the percentage of cells expressing the protein and the intensity of the signal in over 10,000 cells per knockdown model. Taking both metrics into account, we confirmed that the effects observed on

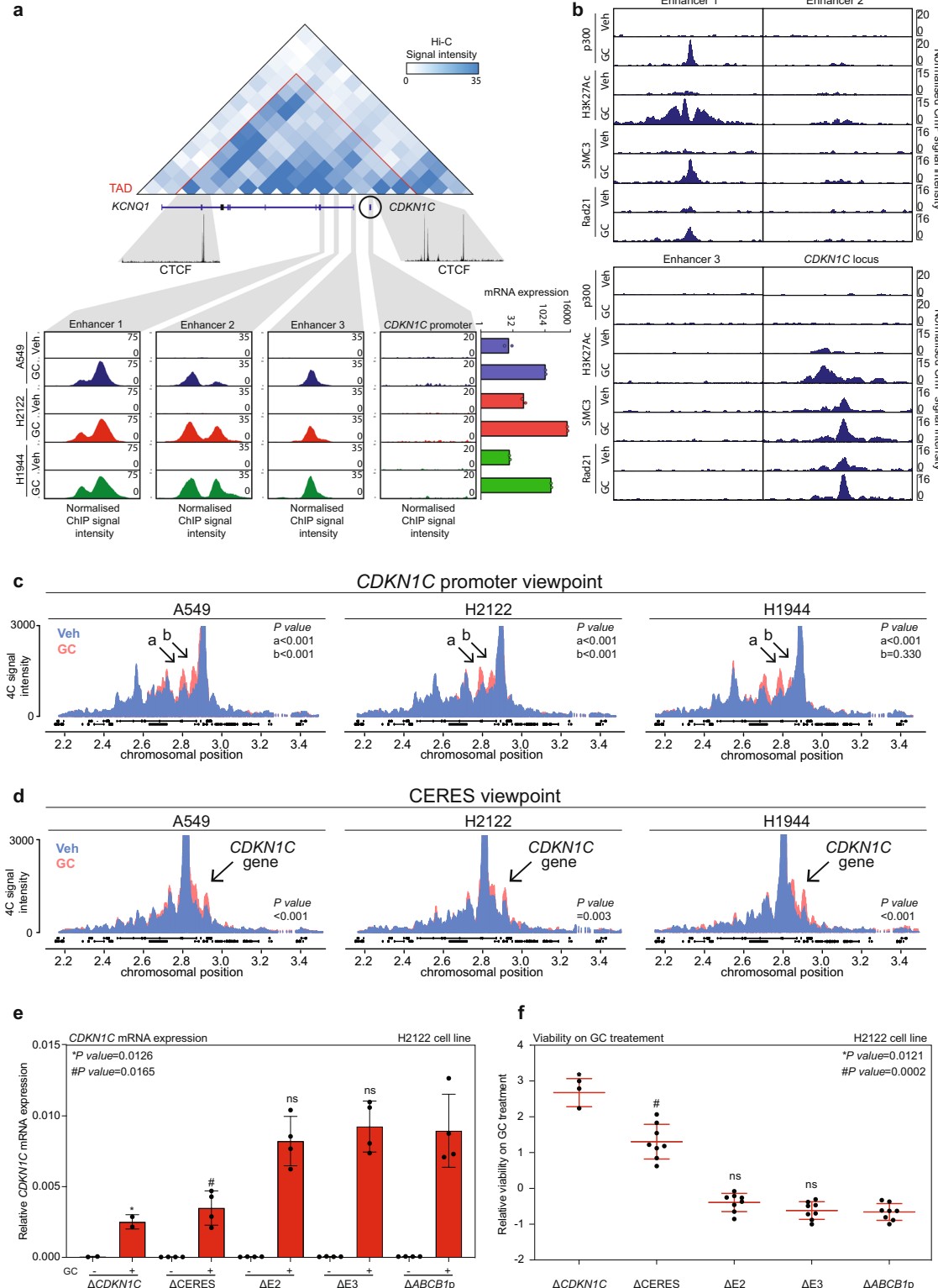

transcription level are also seen on protein level (Fig. 5h, right), further strengthening our conclusions that GR gene regulation and dormancy induction are under the direct control of SWI/SNF complex functionality and composition.

With this, we have shown that the SWI/SNF remodelling complex forms an essential part of the GR transcriptional machinery, necessary for the regulation of p57, which is required to drive cells into dormancy.

**Accessibility of CERES in human cancer samples is associated with GR-dependent *CDKN1C* expression and activity.** To investigate the clinical validity of the GR-driven *CDKN1C* enhancer identified in this study, we explored transcriptomics and chromatin accessibility (Assay for Transposase-Accessible Chromatin (ATAC) sequencing) datasets of the TCGA cohort (Fig. 6a). We observed that samples with high chromatin accessibility of CERES have high expression of *CDKN1C*, accompanied

**Fig. 4 GR-bound enhancer within *KCNQ1* gene regulates *CDKN1C* through chromatin looping. a** (top) Hi-C contact map of A549 cell line at the resolution of 40 kb of the region surrounding *CDKN1C* gene (chr11:2421230-3141230). CTCF ChIP data (U01HG00790) are represented for TAD anchor sites. **a** (bottom left) GR ChIP sequencing data showing peaks at Enhancers 1, 2 and 3 (chr11:2799339-2801363; chr11:2846111-2848121; chr11:2880971-2882981, respectively), and *CDKN1C* promoter (chr11:2900697-2910745) for all the cell lines, not-treated (Veh) or treated (GC). **a** (bottom right) mRNA expression of *CDKN1C* with or without glucocorticoids (n = 2). **b** Normalised ChIP signal for p300 (ENCSR571KWZ), H3K27Ac (ENCSR375BQN), SMC3 (ENCSR376GQA) and Rad21 (ENCSR501UJL) at Enhancers 1, 2, 3 and *CDKN1C* locus in A549 cell line, untreated (Veh) or glucocorticoid-treated (GC). **c** *CDKN1C* promoter viewpoint 4 C signal across the surrounding region (chromosome 11), under the vehicle (Veh; blue) or glucocorticoid (GC; red) treatment (n = 2). P values were determined by the Wilcoxon t test. **d** Enhancer 1 promoter viewpoint 4 C signal across the surrounding region (chromosome 11), under the vehicle (Veh = blue) or glucocorticoid (GC = red) treatment (n = 2). P values were determined by the Wilcoxon t test. **e** Relative (to the geometric mean of housekeeping genes) *CDKN1C* mRNA expression level of vehicle (−) and GC (+) treated ΔCDKN1C, ΔCERES, ΔE2, ΔE3 and ΔABCB1p cell lines. Mean values ± SD depicted. Two guide RNA pairs per gene (except for ΔCDKN1C; 1 pair was used) in biological duplicates (n = 2). P values were determined by two-sided Welch's t test. **f** Relative viability on GC treatment of ΔCDKN1C, ΔCERES, ΔE2, ΔE3, and ΔABCB1p cell lines. Mean values ± SD depicted. Two guide RNA pairs per genomic location (except for ΔCDKN1C; 1 pair was used) in biological quadruplicates (n = 4). P values were determined by two-sided Mann–Whitney U test.

by low expression of genes involved in cancer cell proliferation (*MKI67* and *PCNA*) and aggressiveness (*FOXM1*) (Fig. 6b). Conversely, samples with low chromatin accessibility of CERES have low expression of *CDKN1C*, and high expression of *MKI67*, *PCNA* and *FOXM1* (Fig. 6b). The observed correlation of chromatin accessibility of CERES and *CDKN1C* expression is significantly higher than the level of correlation seen for any of the 306 enhancers found in the genomic vicinity of *CDKN1C*, which do not correlate with its levels (Fig. 6c). In addition, CERES accessibility does not correlate with the expression of any of the four neighbouring genes proximal to the *CDKN1C* locus (Fig. 6d). Furthermore, correlation between CERES accessibility and *CDKN1C* expression was found to be dependent on GR mRNA levels and increased in a step-wise manner with the removal of tumour samples with the lowest GR expression levels (Fig. 6e). This was not the case for the correlation of accessibility of 306 surrounding enhancers with *CDKN1C* levels (Fig. 6e).

These data support our in vitro findings, suggesting the relevance of GR-mediated regulation of *CDKN1C* by CERES in clinical samples of human non-lymphoid solid cancers.

## Discussion

Pharmacological activation of the GR is a proven, effective treatment strategy for lymphoid cancers, including acute lymphoblastic leukaemia, chronic lymphocytic leukaemia and multiple myeloma[41]. On the other hand, in non-lymphoid solid cancers, GR agonists are commonly prescribed to alleviate the side effects of treatment[41,42]. However, several lines of evidence point towards a direct effect of GR activation on cancer cell behaviour, including invasion[43], apoptosis resistance[44] and growth[7,9,13,45,46]. For example, using in vitro and mouse models of non-small cell lung cancer, it has been shown that GC treatment diminishes cancer incidence and growth[8,10–12,42]. More importantly, several large population-based studies have found that the use of inhaled corticosteroids may reduce the risk of lung cancer development[47–53]. Despite these observations, little is known about the mode of GR action and its direct effectors in non-lymphoid solid cancer types. Therefore, using a multi-disciplinary approach we studied the molecular mechanisms of stress hormone receptor action in non-lymphoid solid cancers, focusing on lung cancer (Fig. 7).

We have found that lung cancer cells react to GCs by transitioning to a dormant state accompanied by activation of IGF-1R survival signalling. While this is the first report of cell dormancy induction by stress hormone receptor activation in cancer, it was previously suggested that GCs may induce cell cycle exit in tenocytes[54], thymic epithelial cells[55], and neural stem cells[56], potentially contributing to long-term degenerative changes in tendon tissue, development of neural disorders and T cell-

mediated autoimmune diseases, respectively. It is highly likely, however, that cellular and molecular modes of dormancy activation may differ between target tissues and/or pathologies due to the cell-type-specific nature of GR action[3,57].

Stress hormone-induced cell dormancy is driven by p57 and is characterised by both attributes independently associated with either senescence (senescence-associated β-galactosidase positivity and enrichment of a senescence-related gene signature) or quiescence (reversible state, a silent metabolic profile, lack of p53 response and p16/p21 upregulation, and activation of growth factor signalling (IGF-1R)) according to the recent guidelines by the International Cell Senescence Association[58]. The initiating driver of the phenotype reported in this manuscript is p57, a cell cycle inhibitor, known to have additional mechanisms[59,60] in comparison to the other family members that drive quiescence (p27) and senescence (p16/p21). Recent evidence exists of p57 being able to initiate both senescence and quiescence in human primary tissue models depending on environmental cues[61]. In relation to this, it could be hypothesised that the composite phenotype we observed (in part senescence, in part quiescence) is caused by the altered action of p57 in cancer. In contrast to prior studies on cancer models[62–68], the cell cycle exit and dormant state reported in this manuscript is induced by a physiological ligand found in circulation, to which all cancers are exposed, and is not caused by pharmacologically induced DNA damage or inhibition of cell cycle machinery.

Of particular interest is the GC-induced tolerance to various anticancer drugs. While the decrease in sensitivity to selected chemotherapeutics after GC treatment has been observed previously[44,69–72], we have unbiasedly profiled a large number of compounds to show that this generally applies to various drugs. Our findings complement previous reports[44,69–72] in raising concern about the widespread use of GCs in the management of anticancer therapy side effects and warrant clinical caution and investigation. The viability in this GC-induced multidrug-tolerant state is maintained via engagement of IGF-1R signalling. As this signalling plays a significant role in maintaining cell survival[73,74], the induced activity of this pathway might provide key survival mechanisms, potentially yielding a therapeutic opportunity, as demonstrated by a significant reduction in viability upon its inhibition. It was previously observed that IGF-1R activity is necessary for cell viability maintenance of cancer cells subpopulations following lethal drug exposure[75]. In addition, activity of this signalling is needed for entry and exit from quiescence in a nutrient-depletion pancreatic cancer in vitro model[76]. In conjunction with this, it could be hypothesised that IGF-1R activation in accordance with the circulating levels of GCs and the circadian rhythm[77] may also enable cancer cells to readily react to

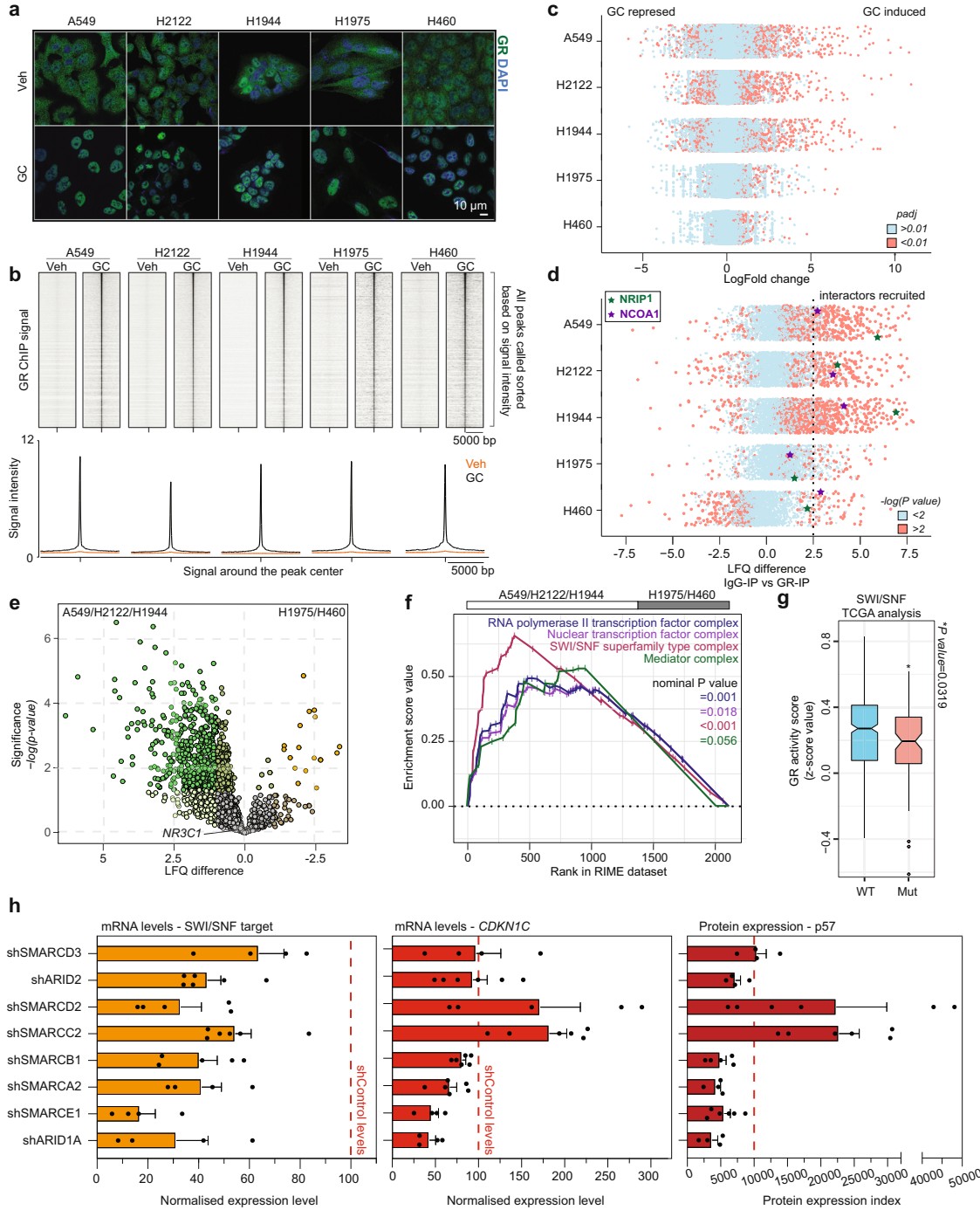

daily changes in the environment, allowing them to rapidly switch between a proliferating state and dormancy.

It is well documented that GR activation modulates the expression of a large number of genes throughout the genome[3]. However, direct causality in linking particular genes to phenotypes is still understudied with a limited number of examples described to date[78–80]. Our findings demonstrate that a single GR-target gene (p57) drives the induction of the reported dormancy phenotype, as discussed above.

The most complex member of the Cip/Kip family, p57, is a key protein involved in the development, organ morphogenesis, and tumour suppression[31,60,81]. This protein exerts its function through interaction with and direct inhibition of various cyclin-dependent kinases, which in turn leads to dephosphorylation of cell cycle proteins (including retinoblastoma), inactivation of E2F and cell cycle exit[82]. While the molecular mechanisms behind p57-induced cell cycle exit are known[82], the regulation of this key cell cycle inhibitor by transcription factors is proposed to be complex, cell-type specific, and not entirely understood[83]. We show that GR directly regulates CDKN1C gene expression, through induction of chromatin looping initiated by binding to CERES, a distal enhancer located in the KCNQ1 gene, in a SWI/SNF chromatin remodelling complex-dependent fashion.

Precise regulation of CDKN1C is imperative for embryogenesis, differentiation, as well as tumour suppression[31,60,81,82]. In relation to its diverse roles, CDKN1C expression is dynamically linked to

**Fig. 5 SWI/SNF complex fine-tunes expression of *CDKN1C*. a** Representative immunofluorescence images showing expression and localisation of GR (green), treated with glucocorticoids (GC) or control (Veh), using DAPI as nuclear staining (blue) (*n* = 3). Scale bar, 10 μm. **b** Heatmap of ChIP-sequencing signal around peak midpoint for all sites detected across the genome (top), and average signal of GR ChIP-seq experiments across all sites called over input control (bottom), for untreated (Veh) and glucocorticoid-treated (GC) cells. **c** Scatter plot depicting differential gene expression changes upon GC treatment in RNA sequencing. Genes significantly (*P*adj ≤ 0.01) up- or downregulated by GCs are depicted in red (*n* = 2). Adjusted *P* values were determined by DESeq2 (Wald test *P* values corrected for multiple testing using Benjamini and Hochberg method). **d** Scatter plot depicting enrichment over IgG control in a GR-RIME experiment. Proteins considered to be recruited by GR are 2.5 LFQ enriched over IgG (dotted line) and significant (−log(*P* value) >2; red) (*n* = 3). *P* values were determined by two-sided *t* test. **e** Volcano plot depicting differentially enriched interactors in GR-RIME experiments between three cell lines with active and two cell lines with inactive GR (*n* = 3). *P* values were determined by two-sided *t* test. **f** GSEA enrichment profiles for RNA polymerase II transcription factor complex (M17103; blue). Nuclear transcription factor complex (M17532; purple), SWI/SNF complex (M17713; red) and mediator complex (M17759; green) gene sets based on A549/H2122/H1944 and H1975/H460 comparison GR-RIME dataset (*n* = 3). Nominal *P* values were determined by GSEA software. **g** Box plot depicting GR activity (*z* score of 253 genes) in SWI/SNF WT (*n* = 786) and mutant (*n* = 91) human lung adenocarcinoma tumours. The central mark indicates the median, and the bottom and top edges of the box indicate the 25th and 75th percentiles, respectively. The notch displays a confidence interval around the median. The maximum whisker lengths are specified as 1.5 times the interquartile range and outliers are depicted as filled circles. *P* values were determined by Wilcoxon rank-sum test with continuity correction. **h** (left) Normalised mRNA expression level relative to shControl for *SMARCD3, ARID2, SMARCD2, SMARCC2, SMARCB1, SMARCA2, SMARCE1* and *ARID1A*, in cell lines with shRNA targeting respective genes. Mean values with ± SEM depicted. ≥2 shRNAs per gene in biological duplicates (*n* = 2). **h** (middle) Normalised (relative to untreated condition) *CDKN1C* mRNA expression level of GC-treated shSMARCD3, shARID2, shSMARCD2, shSMARCC2, shSMARCB1, shSMARCA2, shSMARCE1 and shARID1A H2122 cell lines. Mean values with ±SEM depicted. ≥2 shRNAs per gene in biological duplicates (*n* = 2). **h** (right) Protein expression index (number of positive cells * average signal intensity) depicting quantified expression of p57 in immunofluorescence experiments using shSMARCD3, shARID2, shSMARCD2, shSMARCC2, shSMARCB1, shSMARCA2, shSMARCE1 and shARID1A H2122 cell lines. Mean values with ±SEM depicted. ≥2 shRNAs per gene in biological duplicates (*n* = 2), ≥10,000 cells quantified.

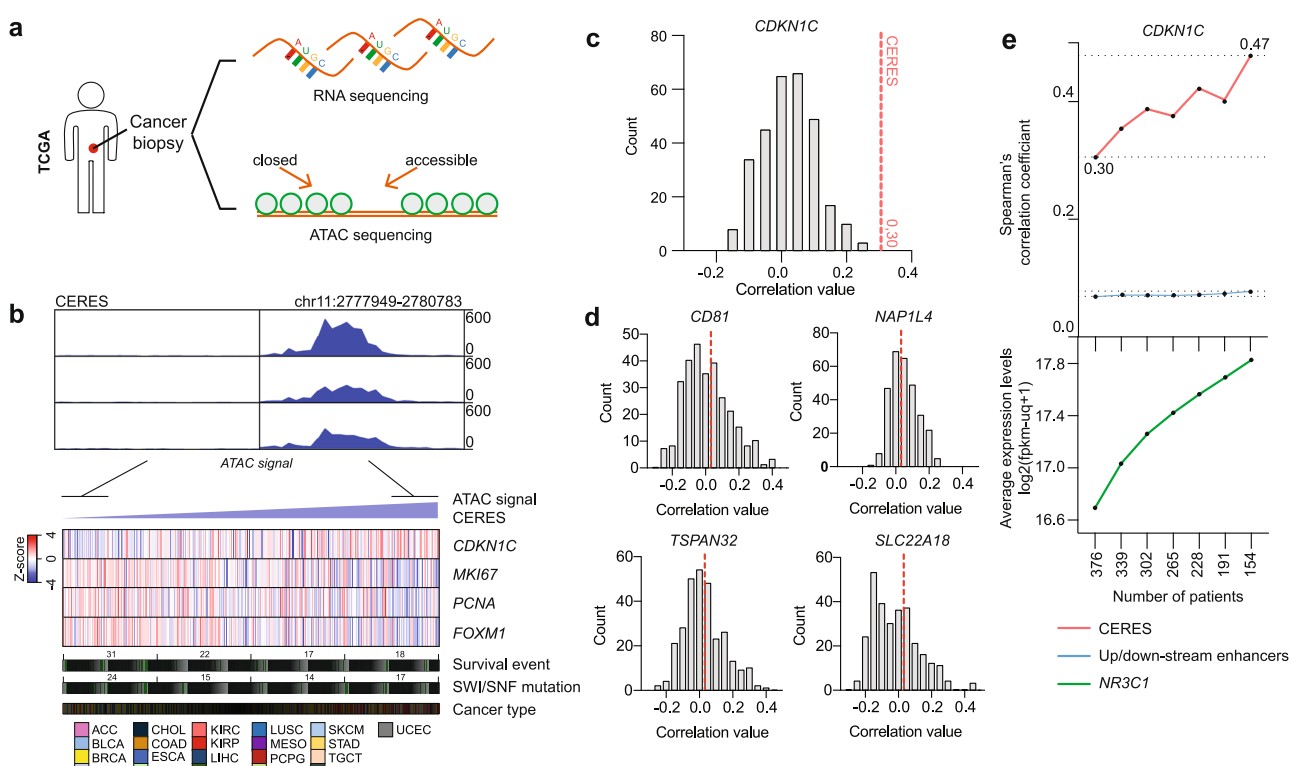

**Fig. 6 *CDKN1C* mRNA expression is linked to chromatin accessibility of CERES and GR levels in clinical samples. a** Biopsies of human primary non-lymphoid solid cancers were subjected to whole-transcriptome and chromatin accessibility analysis by the TCGA (*n* = 404). **b** (top) Representative samples showing chromatin accessibility signal at CERES enhancer. **b** (bottom) Expression of *CDKN1C, MKI67, PCNA*, and *FOXM1* sorted based on CERES accessibility. Survival events (alive = white, dead = green), SWI/SNF mutation status (WT = white, mutated = green), and cancer types are displayed below. **c** Histogram depicting correlation value counts for *CDKN1C* expression levels and 306 up- and downstream enhancers in its genomic vicinity. Red line shows the correlation of CERES accessibility with *CDKN1C* expression. **d** Histograms showing correlation of four proximal genes with the accessibility of 306 up- and downstream enhancers surrounding *CDKN1C*. Red lines show correlations of these genes with CERES accessibility. **e** Line plot showing Spearman's correlation of accessibility of CERES (red) or 306 (blue) surrounding enhancers with *CDKN1C* mRNA levels, and *NR3C1* mRNA levels (green).

numerous homeostatic cues and the circadian rhythm in various tissue types[84]. Prior research has suggested that *CDKN1C* is regulated in a complex combinatorial fashion, through an imprinting control region and unidentified enhancers located in the *KCNQ1* gene[33,34].

In this study, we describe a GR-driven distal enhancer (>100 kb upstream of *CDKN1C*, within an intron of *KCNQ1* gene) of which the physical interaction with the *CDKN1C* promoter is enhanced by GR activation. Importantly, we functionally probed this enhancer

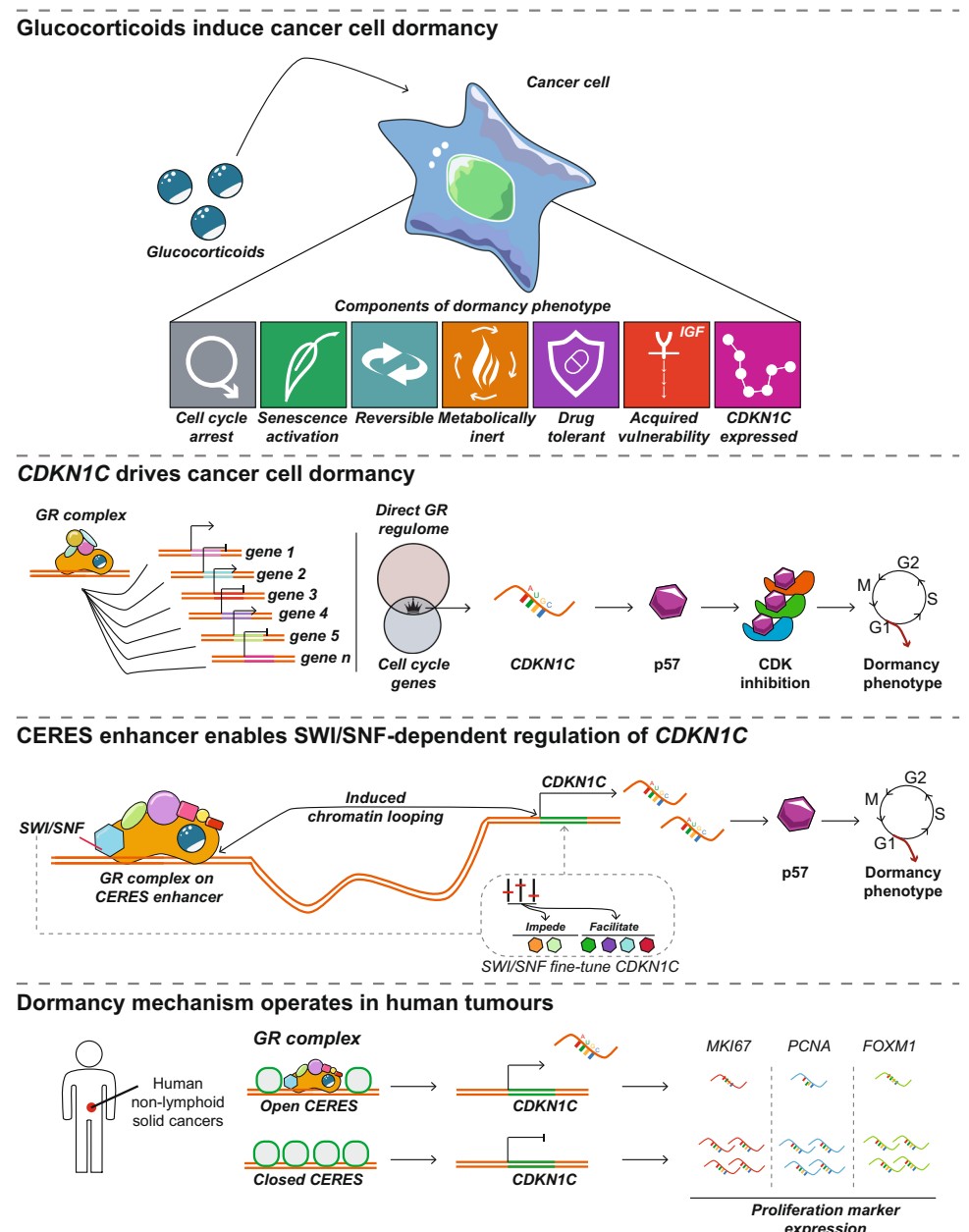

**Fig. 7 Mechanism of GR-driven, reversible drug-tolerant cancer cell dormancy.** Glucocorticoids act on lung cancer cells by evoking a reversible cell cycle arrest, activating senescence programs, attenuating the metabolic processes, reducing sensitivity to a large array of drugs, inducing dependency towards growth hormone signalling (IGF-1R) and initiating expression of *CDKN1C*. While GR regulates a large number of genes throughout the genome, we identified that the cell dormancy phenotype is driven by a single GR-target gene, *CDKN1C*, which encodes for p57, a potent CDK inhibitor protein. This GR target is necessary for the attainment of the cell dormancy phenotype. Regulation of *CDKN1C* involves a distal enhancer CERES (*C*DKN1C Enhancer Regulated by Steroids) located upstream of *CDKN1C* in the *KCNQ1* gene, through a GR-induced chromatin looping event. GR and its transcription machinery initiate transcription from the *CDKN1C* locus, this in turn leads to the acquisition of a cell dormancy phenotype. Crucial for transcriptional activity of GR at *CDKN1C* locus is the SWI/SNF complex, as its members fine-tune GR transcriptional output allowing for precise control of gene expression. This mechanism of regulation is observed in non-lymphoid solid human cancers, where it contributes to tumour suppression.

using CRISPR-Cas9-based deletion to demonstrate that this enhancer is critically involved in GR-driven *CDKN1C* expression. We also presented evidence that GR-mediated regulation of *CDKN1C* by the CERES enhancer occurs in clinical samples, highlighting the relevance of this regulation in tumour development and progression. Strikingly, a deletion in the *KCNQ1* gene spanning the discovered genetic element CERES was detected and causally linked to Beckwith–Wiedemann Syndrome and Silver–Russell Syndrome, both known to be consequences of *CDKN1C* loss/downregulation[85,86],

suggesting a physiological role of the discovered enhancer in maintaining expression of this important cell cycle regulator.

Our GR transcriptional complex analysis gives insight into how this factor operates in cancer. We show that upon chromatin binding, GR recruits a proficient transcriptional complex composed of coregulators, SWI/SNF chromatin remodelers, mediator subunits and the RNA polymerase machinery. Specifically, as SWI/SNF members have been related to GC-resistance in clinical models of acute lymphoblastic leukaemia, we experimentally

probe and demonstrate that p57 regulation by GR is SWI/SNF-dependent. The members of the SWI/SNF family have been found to be involved in gene regulation[87–89], mostly through activating transcription, however, they may also directly repress gene expression[90]. In terms of GR biology, prior work has defined SWI/SNF, especially the BRG1 subunit (SMARCA4) as a critical component promoting transcriptional activity of this nuclear receptor[91–94]. Our findings, albeit limited to the key driver of cell dormancy, show that the role of SWI/SNF members in GR biology is more complex than originally thought. Specifically, while some SWI/SNF subunits seem to be involved in the positive regulation of CDKN1C, others seem to repress GR transcription for this gene. To our knowledge, this is the first report showing that particular subunits of SWI/SNF complex antagonise GR transcriptional modulation. Overall our findings suggest that the specific composition of the SWI/SNF complex may play a role in adjusting the level of transcriptional output of GR at the CDKN1C locus.

We speculate that the role of GR to induce cellular dormancy in cancer may be reminiscent of its signalling in normal tissue. In agreement with this, activation of GR has previously been linked to cell differentiation and lineage selection[95,96]. As tumours are exposed to GCs produced by the adrenal gland and released into the circulation, this dormant state might be a feature of various early-stage human tumours, supported by the observation that GR expression is lower in various cancer types (including lung, breast and prostate) as compared to normal tissue and that it may serve as a tumour suppressor[14]. Furthermore, cell dormancy has been shown to be under circadian control in various tissues and stem cells[97,98]. In relation to that, the well-known day–night rhythmic behaviour of GC levels may also suggest that cell dormancy, as well as chemotherapy response in cancer, is subjected to circadian rhythm through intratumoral GR activity.

We conclude that SWI/SNF-dependent expression of CDKN1C, facilitated by looping of a specific GR-bound distal enhancer CERES, induces a reversible dormancy state in which cells become tolerant to a large array of anticancer drugs and acquire IGF-1R signalling dependency.

## Methods

**Cell lines**. A549, H2122, H1944, H1975 and H460 cells were obtained from Rene Bernard's lab (Netherlands Cancer Institute, Netherlands). HEK293T cells were obtained from American Type Culture Collection (ATCC). The human mesothelioma cell lines M28 and VAMT were provided by Courtney Broaddus (University of California, USA), while the NCI-H2795 (H2795) was obtained from Ultan McDermott (Sanger Institute, UK). The primary human mesothelioma cell lines PV130913, PV041214, PV020318, PV150318, PV240418, PV180518 and PV250518 were generated by Laurel Schunselaar (Netherlands Cancer Institute, Netherlands).

A549 and H2795 cell lines were maintained in DMEM/F12 (1:1) (1X) + Glutamax (Life Technologies), while HEK293T, M28, PV130913, PV041214, PV020318, PV150318, PV240418, PV180518 and PV250518 were cultured in DMEM, both media supplemented with 10% foetal calf serum (FCS). The other cell lines were cultured in RPMI-1640 (phenol-red + glutamine) (Life Technologies) medium supplemented with 10% FCS. All the cell lines were authenticated and tested negative for mycoplasma contamination.

For hormone deprivation, cells were cultured in medium containing 5% charcoal-treated FCS for 3 days, subsequently treated with 2.75 μM hydrocortisone (Sigma) or vehicle, and harvested at the indicated timepoint. Specific conditions, treatment duration and compounds used can be found in Table S1.

**Generation of CRISPR cell lines**. Guide RNAs targeting human CDKN1C (CTGGTCCTCGGCGGTTCAGCT), NR3C1 (GTGAGTTGTGGTAACGTTGC) and a non-targeting control guide (GTATTACTGATATTGGTGGG) were individually cloned into the lentiCRISPR v2 plasmid[99]. CRISPR vectors were co-expressed with 3rd generation viral vectors in HEK293T cells using polyethyleneimine (PEI). After lentivirus production, the medium was harvested and transferred to the designated cell lines. Two days post infection, cells were put on puromycin selection for 3 weeks.

For genomic excision experiments, guides were designed using DeepSpCas9[100] and cloned into lentiCRISPR v2 plasmid. For each locus, two pairs (except for

CDKN1C gene) of guides were used: CERES—pair 1: GGCACCCTGGGGAGCTG CCGG and TGGAGTCTGGCATCCTGCGG, pair 2: AGCAGACTACGAGAGG CAGC and ACCGACACTCCTGCACGTCA; E2—pair 1: TGCAGACCCAGAC CAATGGG and GGGCAGGGGCCCCAAGACGA, pair 2: TGGATGGAAGCTG TACGATG and ACTGTCTGCGCGGAACCGCA; E3 promoter—pair 1: GCAGC TGCCACAGCATGCAG and TGAGCAGCCCAACGGCATGG, pair 2: GGTCTG CTGGAAGCCAATGC and ACCGTAGCTGGACCTTCACCC; ABCB1 promoter— pair 1: GGTGAATGACTAAGAACGGT and GCCGCTACTCGAATGAGCTC, pair 2: AGCAGCATATGGCTCACGTG and ACAGATGACTGCTCCCGGCC; and CDKN1C – selected from GeCKOv2 human CRISPR knockout library. CRISPR vectors were co-expressed with 3rd generation viral vectors in HEK293T cells using polyethyleneimine (PEI). After lentivirus production, the medium was harvested and transferred to the designated cell lines. Two days post infection, cells were put on puromycin selection for 3 weeks.

**shRNA experiments**. shRNA knockdown experiments were carried out by infection of H2122 cells with pLKO.1-puro containing a non-targeting, SMARCD3-, ARID2-, SMARCD2-, SMARCC2-, SMARCE1-, SMARCA2-, SMARCB1- or ARID1A-specific hairpin obtained from The RNAi Consortium (TRC) library (https://www.broadinstitute.org/rnai-consortium/rnai-consortium-shrna-library).

**RNA sequencing**. Cells were serum-starved for 3 days before they were treated with hydrocortisone (2.75 μM) for 8 h. Total RNA was isolated using the RNeasy Mini Kit (Qiagen, Germany) according to the manufacturer's instructions. The quality and quantity of the total RNA were assessed by the 2100 Bioanalyzer using a Nanochip (Agilent, USA). Total RNA samples having an RNA integrity number (RIN) above 8 were subjected to library generation.

Strand-specific libraries were generated with the TruSeq Stranded mRNA sample preparation kit (Illumina, Part # 15031047 Rev. E) and sequenced on a HiSeq2500. RNA sequencing data were mapped to exons using Tophat (v.2.1). Read counting, normalisation and differential gene expression were performed using R package DESeq2[101]. Gene-set enrichment analysis and gene-list analysis were executed according to the instructions[102,103].

**ChIP-sequencing**. Chromatin immunoprecipitations were performed as previously described[104]. Nuclear lysates were incubated with 7.5 μl of GR antibody (D6H2L, Cell Signalling Technology) pre-bound to 50 μl of protein A beads per samples. Immunoprecipitated DNA was processed for library preparation (0801-0303, KAPA biosystems kit). Samples were sequenced using an Illumina Hiseq2500 genome analyser (65 bp reads, single end), and aligned to the Human Reference Genome (hg19, February 2009). Reads were filtered based on MAPQ quality ((samtools v1.8); quality ≥20) and duplicate reads were removed (Picard Mark-Dupes v2.18). Peak calling over input control was performed using MACS2 (v2.1.1) peak caller. MACS2 was run with the default parameters. Genome browser snapshots, heatmaps and density plots were generated using EaSeq (http://easeq.net)[105].

**Western blot**. Cells were lysed in 2× Laemmli buffer (120 mM Tris, 20% glycerol, 4% SDS). Total protein content was quantified by BCA assay (23227, Thermo Fisher Scientific). Cell lysates containing equal amounts of protein were analysed by SDS-PAGE, after protein transfer, nitrocellulose membranes were incubated with antibodies against GR (12041, Cell Signalling Technology, 1:1000), ER (MA5 14104, Thermo Fisher Scientific, USA, 1:1000), AR (#06680, Merck Millipore, USA, 1:1000), PR (sc-539, Santa Cruz Biotechnology, 1:1000), p57 (sc-56341, Santa Cruz Biotechnology, 1:500), PARP (9542, Cell Signalling Technology, 1:1000), p53 (sc-126, Santa Cruz Biotechnology, 1:1000), Actin (MAB1501R, Merck, 1:10,000) or Hsp90 (sc-13119, Santa Cruz Biotechnologies, 1:1000).

**Seahorse**. Cellular respiration was measured using a Seahorse XF24 Bioanalyzer (Seahorse Biosciences). A549, H2122, and H1944 cells were all seeded at 75,000 cells per well to XFe24 cell-culture microplates (102340-100, Seahorse Biosciences) and cultured overnight before the analysis. The analysis was performed according to the manufacturer's instructions in DMEM (D5030, Sigma-Aldrich) supplemented with 10 mM D-glucose and 4 mM L-glutamine for the oxygen consumption rate (OCR) experiments. For OCR measurements, the following reagents that selectively inhibit mitochondrial function[106] were added: oligomycin (1 μM; an ATP synthase (complex V) inhibitor), FCCP (0.4 μM; an uncoupling agent that collapses the proton gradient and disrupts the mitochondrial membrane potential), and rotenone (1 μM; a mitochondrial complex I inhibitor) and antimycin A (1 μM; a mitochondrial complex III inhibitor). Results were normalised to DNA content using nanodrop quantification (Thermo Fisher Scientific).

**Senescence-associated β-galactosidase assay**. Cytochemical staining for senescence-associated β-galactosidase was performed as described before[107]. Cells were serum-starved for 3 days and subsequently fresh medium with or without hydrocortisone (2.75 μM) was added for 2 days. After incubation, cells were washed twice with PBS and fixed with 3.7% formaldehyde for 5 min. Following that, cells

were washed with PBS before they were incubated with X-gal staining solution (1 mg/mL X-gal, 40 mM citric acid/sodium phosphate buffer, 5 mM potassium ferricyanide, 5 mM potassium ferrocyanide, 2 mM MgCl$_2$, 150 mM NaCl) overnight at 37 °C. X-gal is an artificial substrate of the β-galactosidase enzyme and is used to detect senescence-associated β-galactosidase. The next day, cells were washed with PBS and imaged with a Zeiss Axiovert S100 inverted microscope (Zeiss, Germany).

**Cell cycle analysis with flow cytometry.** Cells were serum-starved for at least 3 days and subsequently, either treated with hydrocortisone (2.75 μM) or untreated (FCS) for 2 days. Subsequently, cells were harvested and centrifuged for 4 min at 335×g at 4 °C. The pellet was resuspended in cold PBS and cells were fixed in cold 80% EtOH overnight at 20 °C. The next day, cells were centrifuged for five minutes at 425×g at 10 °C and incubated with 500 μL of PI staining mix (50 μg/mL PI, 100 μg/mL RNAse A, 0.5% Triton™ X 100) for 40 min at 37 °C. Following that, PBS was added, and samples were centrifuged for 5 min at 425×g at 4 °C. Pellets were resuspended in PBS and stored at 4 °C till flow cytometric analyses. Flow cytometric measurements were performed on LSRFortessa SORP 2 (BD Biosciences) and cell cycle distribution analysed with FlowJo Software (FlowJo LLC).

**Annexin V/propidium iodide apoptosis assay for flow cytometry.** Annexin V/ propidium iodide apoptosis assays were performed as described before[108]. Cells were serum-starved for at least 3 days. Following that, cells were either left untreated (FCS) or treated with HC at a concentration of 2.75 μM for 6 days. As a positive control, cells were treated for ~30 h with 50 μM cisplatin to induce apoptosis. Cells were harvested and centrifuged for 10 min at 335×g at 4 °C. The pellet was washed twice, once with cold PBS and a second time with Annexin V binding buffer (10 mM HEPES, 140 mM NaCl, 2.5 mM CaCl$_2$, pH 7.4). After centrifugation for 10 min at 335×g at 4 °C, cells were resuspended in Annexin V binding buffer and Annexin V (Thermo Fisher Scientific, USA) was added according to the manufacturer's recommendations. Samples were incubated for 15 min at room temperature in the dark before propidium iodide (PI) was added at a concentration of 2 μg/mL. Following an additional incubation for 15 min at RT, cells were washed with Annexin V binding buffer and centrifuged for 10 min at 335×g at 4 °C. Cells were resuspended in Annexin V binding buffer with 1% formaldehyde and fixed for 10 min on ice or overnight at 4 °C. Subsequently, cells were washed twice with cold PBS and centrifuged for 8 min at 425×g at 4 °C. The pellets were resuspended in Annexin V binding buffer and RNase A (50 μg/mL; Thermo Fisher Scientific, USA) was added. Samples were incubated for 15 min at 37 °C and washed once more with cold PBS. Afterwards, cells were centrifuged for 8 min at 425×g at 4 °C, resuspended in PBS and stored at 4 °C till flow cytometric analyses. Flow cytometric measurements were performed on an Attune NxT Flow Cytometer (Thermo Fisher Scientific, USA) and cell populations analysed with FlowJo Software (FlowJo LLC, USA).

**Xenografts.** The H1944 or A549 cells were trypsinised and resuspended in PBS at a density of six (H1944) or two (A549) million cells/50 μl and mixed with an equal volume of BME (#3533-005-02, Sigma-Aldrich) NOD-scid-γ (NSG) mice (±7 weeks old) were anesthetised before injection of six million cells subcutaneously into one of the flanks. Once the tumour size reached between 100 and 300 mm$^3$ the mice were treated with 4 mg/kg dexamethasone (D2915-100MG, Sigma-Aldrich; dissolved in water), 75 (H1944) and 20 (A549) mg/kg Linsitinib (HY-10191, MedChemExpress; dissolved in 25 mM tartaric acid), 25 mg/kg GSK1838705A (MedChemExpress; dissolved in 20% sulfobutyl ether β-cyclodextrin (ISP; pH 3.5)), combination, or vehicle by I.P. injections (dexamethasone) and orally (IGF-1R inhibitors) on a daily basis. Tumour volume was monitored by calliper measurements every 2 days. Mice were kept under standard temperature and humidity conditions in individually ventilated cages, with food and water provided ad libitum. The NKI Animal Experiments Committee approved all in vivo experiments (project number 9139 and 9907).

**RNA isolation and mRNA expression.** Total RNA was isolated using TRIzol Reagent (Thermo Fisher Scientific, USA), and cDNA was synthesised from 2 μg RNA using the SuperScript™ III Reverse Transcriptase system (Thermo Fisher Scientific, USA) with random hexamer primers according to the instructions provided by manufacturers. Quantitative PCR (qPCR) was performed using the SensiMix™ SYBR Kit (Bioline, UK) according to the manufacturer's instructions on a QuantStudio™ 6 Flex System (Thermo Fisher Scientific, USA). Primer sequences for mRNA expression analysis are listed in Supplementary Table 2.

**Rapid immunoprecipitation of endogenous proteins (RIME).** Following hormone deprivation and treatment with hydrocortisone (2.75 μM) for 2 h, RIME experiments were performed as previously described[32]. The following antibodies were used: anti-GR (12041, Cell Signalling Technology), anti-p57 (sc-56341, Santa Cruz Biotechnology), anti-rabbit IgG (sc-2027, Santa Cruz Biotechnology) and anti-mouse IgG (sc-2025, Santa Cruz Biotechnology).

Tryptic digestion of bead-bound proteins was performed as described previously[109]. LC-MS/MS analysis of the tryptic digests was performed on an Orbitrap Fusion Tribrid mass spectrometer equipped with a Proxeon nLC1000 system (Thermo Scientific) using the same settings, with the exception

that the samples were eluted from the analytical column in a 90-min linear gradient.

Raw data were analysed by Proteome Discoverer (PD) (v. 2.3.0.523, Thermo Scientific) using standard settings. MS/MS data were searched against the Swissprot database (released 2018_06) using Mascot (v. 2.6.1, Matrix Science, UK) with Homo sapiens as taxonomy filter (20,381 entries) for the GR-RIME experiment, whereas Sequest HT was used for the p57-RIME experiment. The maximum allowed precursor mass tolerance was 50 ppm and 0.6 Da for fragment ion masses. Trypsin was chosen as cleavage specificity allowing two missed cleavages. Carbamidomethylation (C) was set as a fixed modification, while oxidation (M) and deamidation (NQ) were used as variable modifications. False discovery rates for peptide and protein identification were set to 1% and as an additional filter Mascot peptide ion score >20 or Sequest HT XCorr>1 was set. The PD output file containing the abundances was loaded into Perseus (version 1.6.1.3) [02] LFQ intensities were Log2-transformed and the proteins were filtered for at least 66% valid values. Missing values were replaced by imputation based on the standard settings of Perseus, i.e., a normal distribution using a width of 0.3 and a downshift of 1.8. Differentially expressed proteins were determined using a t test. The comparison between the pooled cell lines of the GR-RIME experiment was IgG corrected.

**Immunofluorescence and quantification.** After hormone deprivation cells were treated with 2.5 μM hydrocortisone or left untreated for 8 h. Cells were washed and fixed in 2% paraformaldehyde for 10 min at room temperature. Subsequently, cells were permeabilized in 0.5% Triton-PBS for 10 min. After blocking for 60 min with blocking solution (1% BSA in PBS), samples were incubated for 2 h with antibodies against GR (1:50), p57 (1:50) at room temperature. Following that, samples were incubated with secondary antibodies: Goat anti-Mouse IgG (H + L) Cross-Adsorbed Secondary Antibody, Alexa Fluor 488 (A11001, Thermo Fisher Scientific) (1:1000) and goat anti-rabbit IgG (H + L) Cross-Adsorbed Antibody, Alexa Fluor 647 (A21244, Thermo Fisher Scientific) (1:1000) Finally, samples were counterstained with 4′,6-diamidino-2-phenylindole (DAPI) and analysed by either laser confocal microscopy (SP5, Leica) or screening fluorescent microscope (TIRF, Leica).

For single-cell analysis, images were analysed in FIJI[110]. p57-positive cells were quantified in a fully automatic, unbiased manner, with a custom-made ImageJ macro script. For every image, the DAPI channel was used to segment cell nuclei into ROIs as follows. After rolling ball background subtraction (40-micron radius) and a median filter (1.5-micron radius), local thresholding was applied ('Mean' method, 8-micron radius, with four times the standard deviation of the background as a parameter), followed by a distance transform watershed operation to separate touching nuclei. The mean p57 signal was then measured inside the obtained ROIs. Cells were considered to be positive (negative) if this mean value was higher (lower) than a certain threshold, determined using untreated control samples. The resulting images with filled ROIs were overlaid with the original data for visual inspection.

**Drug screen.** Before the start of the screen, H1944 cells were cultured in medium without or with glucocorticoids (hydrocortisone, 2.75 μM) for 2 days. Using the Multidrop Combi (Thermo Fisher Scientific), untreated H1944 cells were seeded into 384-well plates either at low (1000 cells) or high (2500 cells) confluency, while the pre-treated H1944 cells were seeded at high (4500 cells) confluency. After 24 h, the NKI compound collection of purchased drugs (Selleck GPCR (256 drugs), Kinase inhibitors (411 drugs), apoptosis targets (23 drugs), phosphatase inhibitors (33 drugs), epigenetic inhibitors (160 drugs), LOPAC (1280 drugs) and NCI oncology (114 drugs)) was added. This library was stored and handled as recommended by the manufacturer. Compounds from the master plate were diluted in daughter plates containing complete RPMI-1640 medium, using the MICROLAB STAR liquid handling workstation (Hamilton). From the daughter plates, the diluted compounds were transferred into 384-well assay plates, in triplicate, with final concentrations of 1 μM and 5 μM. In addition, positive (1 μM Phenylarsine oxide) and negative (0.1% DMSO) controls were added alternately to wells in column 2 and 23 of each assay plate. After 6 days, viability was measured using CellTiter-Blue assay (G8081/2, Promega) following the protocol of the manufacturer. The CTB data were normalised per plate using the normalised percentage inhibition (NPI) method. NPI sets the mean of the positive control value to 0 and mean of the negative control to 1. When comparing GC pre-treated vs vehicle and GC-co-treatment vs vehicle, the mean over the three replicates for each condition was calculated and then the vehicle mean was subtracted from the treated condition mean, producing the differential survival value. Using the replicate values of both conditions a two-sided t test was performed. Afterwards, the P values were corrected for multiple testing using the Benjamini–Hochberg method[111]. All calculations were done in R. The same procedure was repeated for H2122 cell lines and was focused on a smaller number of compounds selected on the basis of the first screen (compounds with normalised viability <0.7 in the vehicle arm of H1944 screen treated with 5 μM of library drugs).

**Phosphoproteomic analysis.** After hormone deprivation, cells were treated with 2.75 μM hydrocortisone or left untreated (Veh) for 48 h. For protein digestion,

frozen cell pellets were lysed in boiling Guanidine (GuHCl) lysis buffer as previously described[112]. Protein concentration was determined with a Pierce Coomassie (Bradford) Protein Assay Kit (Thermo Scientific), according to the manufacturer's instructions. Aliquots corresponding to 1.1 mg of protein were digested with Lys-C (Wako) for 2 h at 37 °C, enzyme/substrate ratio 1:100. The mixture was then diluted to 2 M GuHCl and digested overnight at 37 °C with trypsin (Sigma-Aldrich) in enzyme/substrate ratio 1:100. Digestion was quenched by the addition of TFA (final concentration 1%), after which the peptides were desalted on a Sep-Pak C18 cartridge (Waters, Massachusetts, USA). From the eluates, aliquots were collected for proteome analysis, the remainder being reserved for phosphoproteome analysis. Samples were vacuum dried and stored at −80 °C until LC-MS/MS analysis or phosphopeptide enrichment.

Phosphorylated peptides were enriched from 1 mg of total peptides using High-Select Fe-NTA Phosphopeptide Enrichment Kit (Thermo Scientific), according to the manufacturer's instructions, with the exception that the dried eluates were reconstituted in 15 µl of 2% formic acid.

Prior to mass spectrometry analysis, the peptides used for proteome analysis were reconstituted in 2% formic acid. Peptide mixtures were analysed by nanoLC-MS/MS on an Q Exactive HF-X Hybrid Quadrupole-Orbitrap Mass Spectrometer equipped with an EASY-NLC 1200 system (Thermo Scientific). Samples were directly loaded onto the analytical column (ReproSil-Pur 120 C18-AQ, 1.9 µm, 75 µm × 500 mm, packed in-house). Solvent A was 0.1% formic acid/water and solvent B was 0.1% formic acid/80% acetonitrile. Samples were eluted from the analytical column at a constant flow of 250 nl/min. For single-run proteome analysis, a 4-h gradient was employed containing a linear increase from 7 to 30% solvent B, followed by a 15-min wash, whereas for single-run phosphoproteome analysis, a 2-h linear gradient (from 4 to 22% solvent B, followed by a 15-min wash) was used.

Proteome data was analysed by PD (v. 2.3.0.523, Thermo Scientific) using standard settings. MS/MS data were searched against the human Swissprot database (20417 entries, release 2019_02) using Sequest HT. The maximum allowed precursor mass tolerance was 50 ppm and 0.06 Da for fragment ion masses. Trypsin was chosen as cleavage specificity allowing two missed cleavages. Carbamidomethylation (C) was set as a fixed modification, while oxidation (M) and deamidation (NQ) were used as variable modifications. False discovery rates for peptide and protein identification were set to 1%, and as an additional filter Sequest HT XCorr>1 was set. The PD output file containing the abundances was loaded into Perseus (v. 1.6.1.3) [02]. LFQ intensities were Log2-transformed and the proteins were filtered for at least two out of three valid values in one condition. Missing values were replaced by imputation based on the standard settings of Perseus, i.e., a normal distribution using a width of 0.3 and a downshift of 1.8. Differentially expressed proteins were determined using a $t$ test (threshold: $P \leq 0.05$ and $[x/y] \geq 1.5 \mid [x/y] \leq -1.5$).

Phosphoproteome data were analysed by MaxQuant (v. 1.6.1.0) using standard settings[113]. MS/MS data were searched against the human Swissprot database (20,417 entries, release 2019_02) complemented with a list of common contaminants and concatenated with the reversed version of all sequences. The maximum allowed mass tolerance was 4.5 ppm in the main search and 20 ppm for fragment ion masses. False discovery rates for peptide and protein identification were set to 1%. Trypsin/P was chosen as cleavage specificity allowing two missed cleavages. Carbamidomethylation (C) was set as a fixed modification, while oxidation (M), deamidation (NQ) and phosphorylation (S,T,Y) were used as variable modifications. LFQ intensities were Log2-transformed in Perseus (v. 1.6.5.0), after which the phosphosites were filtered for at least two valid values (out of 3 total) in both conditions. Missing values were replaced by imputation based on a normal distribution using a width of 0.3 and a downshift of 1.8. Differentially regulated phosphosites were determined using a $t$ test. These differential phosphosites were combined with on/off (three out of three total present/missing) phosphosites. For the Cytoscape analysis, the app PhosphoPath was used. Data was loaded into the PhosphoPath plug-in and processed as described in the manual[114].

**Time-lapse microscopy.** For doubling time experiments, a Lionheart FX automated microscope was used. Cells (~10,000 per well) were plated in a 96-well plate and sirDNA77 with or without 2.75 µM hydrocortisone was added an hour before imaging. Growth curves were generated with a time resolution of 4 h for a total time span of 144 h (microscope maintained at 37 °C, 5% CO2 using a ×4 lens and a Sony CCD, 1,25-megapixel camera with two times binning; BioTek). Quantification of cell number was performed by Gen5 software (BioTek). Doubling times were calculated using GraphPad Prism 6 software.

For mitotic entry experiments, cells (~20,000 per well) were grown on Lab-Tek II chambered coverglass (Thermo Scientific). One hour before imaging, 2.75 µM hydrocortisone was added per condition. Images were obtained every 15 min during 60 h using a DeltaVision Elite (applied precision) maintained at 37 °C equipped with a ×10 PLAN Apo S lens (Olympus) and cooled CoolSnap CCD camera. Up to five images were acquired per well and 50 cells per experiment were evaluated. Image analysis was performed using ImageJ software (NIH). The percentage of mitotic entry was determined following cells from the start of the movie until they divided up to 60 h.

**4C analysis.** 4C was performed as previously described[115] with minor modifications[116]. 4C libraries were sequenced on a MiSeq and were analysed with a custom 4C mapping pipeline (https://github.com/deWitLab/4C_mapping). 4C ligation data were mapped to hg19. Normalisation and downstream analysis were done using peakC[117]. Vehicle and GC-treated conditions were compared using the Wilcoxon test for the following genetic locations: region a (chr11: 2773921-2812270), region b (chr11: 2830667-2882981) and CDKN1C gene (chr11: 2893641-2926016). Primer sequences are listed in Supplementary Table 2.

**GR ChIP-seq and Hi-C time-course analysis.** Chromatin loops spanning either FOXO1 or IRS2 loci in A549 previously identified using Hi-C[118] were analysed. Dynamic loops showing a significant increase or decrease in chromatin interaction frequency after 1, 4, 8 and 12 h of dexamethasone exposure were reported. GR and c-Jun (AP-1)-binding sites identified by ChIP-seq in a 12 h dexamethasone time course[27] that significantly gained signal and that overlapped a 10 kb anchor of a dynamic loop were interrogated. The changes induced by dexamethasone for GR and c-Jun ChIP-seq signal and chromatin interaction counts were fitted into generalised linear models (GLMs) using edgeR[119] and a likelihood ratio test was performed to identified significant hits (FDR ≤0.05) as previously described. Log2 fold-change values were calculated for each dexamethasone timepoint over the absence of dexamethasone.

**Immunohistochemistry.** Immunocytochemistry xenograft tumour samples were performed by an optimised protocol previously reported[120,121] using the following primary antibodies: Ki67 (ab155580, Abcam), phospho-Rb (Ser780, #9307, Cell Signalling), phospho-IGF-1R (Y1161, ab39398, Abcam), total Rb (ab181616, Abcam), p21 (sc-6246, Santa Cruz), and cleaved caspase-3 (#9661, Cell Signalling).

**Statistical analysis.** Statistical analysis was performed using Prism (GraphPad, San Diego, CA). Normality was tested using D'Agostino–Person and Shapiro–Wilk test. Technique-specific statistical tests are described within their corresponding method subsection.

**Reporting summary.** Further information on research design is available in the Nature Research Reporting Summary linked to this article.

## Data availability

All genomic and mass spectrometry data generated in this study have been deposited in the Gene Expression Omnibus (GEO) and Proteomics Identification (PRIDE) databases, under accession numbers GSE159546 and PXD021924, respectively. Public datasets used in this study are available from GEO or ENCODE, archived under the following codes: U01HG007900, GSE24397 and GSE49591. All the other data are available within the article and its Supplementary Information. Source data are provided with this paper.

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

## Acknowledgements

This work was funded by the Netherlands Organization for Scientific Research NWO VIDI grant 91716401, an Alpe d'Huzes/KWF Bas Mulder Award, KWF grant #12128, and Oncode Institute. Joana Silva was supported by an EMBO long-term fellowship (ALTF 210-2018). Balázs Győrffy was supported by the NVKP_16-1-2016-0037 grant. We would like to acknowledge the NKI-AVL Core Facility Molecular Pathology & Biobanking (CFMPB) for lab support, the NKI Genomics Core Facility for Illumina sequencing and bioinformatics support, and the NKI mouse intervention unit for performing xenograft experiments. We thank Laurel M. Schunselaar for help with mesothelioma pathological samples; Suzan Stelloo for frequent, and helpful discussions; and Sarah Vahed for proofreading.

## Author contributions

S.P. and W.Z. conceived the project outline and coordinated the project. S.P. designed and implemented the project, W.Z. was responsible for project funding and supervision. I.M.P. and A.G.M. have contributed equally to this work. S.P., K.S. and I.M.P. performed RNA and ChIP-sequencing experiments. S.P., K.S. and A.H. performed flow cytometry experiments. A.H. generated all the data on mesothelioma cell lines. S.P. and K.M. evaluated the immunohistochemistry data. S.P., K.S., S.Y. and F.A. generated and performed experiments on knockout/knockdown models. S.P., M.B., D.D., L.H. and M.A. performed mass spectrometry experiments and analysis. A.G.M. performed live-cell imaging experiments and analysis. S.P., K.S., H.T., T.S. and E.dW. performed 4 C

experiments and analysis. A.B. and T.R. analysed GR ChIP and Hi-C time-course data. S. P., J.S. and W.F. performed and analysed seahorse experiments. S.P., K.S., B.M., C.L. and R.B. designed, executed and analysed the drug screen. S.P., M.D.W., J.J. and K.E.dV. designed and analysed xenograft experiments. S.P. and B.vdB. performed image analysis. S.P. performed computational analysis of the majority of the data. S.P., T.C. and B.G. performed computational analysis of TCGA and publically available clinical data. S.P., K. S., I.M.P., A.G.M., R.M. and W.Z. discussed and interpreted the data. S.P. designed and created all the schematics found in the figures. S.P. wrote the manuscript, with input from all authors.

## Competing interests

S.P., K.S. and W.Z. have been designated as inventors in the European patent application #19205735.4 – 1112 filed by Stichting Het Nederlands Kanker Instituut-Antoni Van Leeuwenhoek Ziekenhuis. This patent application is currently pending. The patent application covers the discoveries presented in Fig. 2, S3 and S4. The remaining authors declare no competing interests.

## Additional information

**Peer review information** *Nature Communications* thanks Miles Pufall and the other,

anonymous, reviewer(s) for their contribution to the peer review of this work. Peer reviewer reports are available.

