## [Peer Review File · Nature Communications]

REVIEWER COMMENTS

Reviewer #1 (Remarks to the Author):

The manuscript entitled "Glucocorticoid receptor triggers a novel mode of cell dormancy in solid cancers" explores the effect glucocorticoids on solid cancers, with a particularly deep analysis of non-small cell lung cancer cell lines. Glucocorticoids are a cornerstone in the treatment of lymphoid cancers but have been used in the regimen of prostate cancers, and as an anti-nausea drug to ameliorate the effects of chemotherapies in breast, lung, and other cancers. Their use in these "solid" tumors has been controversial, with some studies showing that glucocorticoids attenuate the effect of chemotherapy on breast cancer tissue, and can even substitute for androgens in castration resistant prostate cancer. A more thorough understanding of how glucocorticoids affect these solid tumor cells is important for understanding their use in the clinic. This study provides important data and an interesting model for how GCs may affect other cancers.

Overall, there is an impressive amount of data in this study on both the cellular phenomena and the mechanism behind them. The authors analyze the data in some useful ways as well. The authors clearly demonstrate that hydrocortisone (which is the same as the endogenous glucocorticoid cortisol) attenuates cell growth in the lung cancer lines without inducing cell death. Glucocorticoids may also have this effect mesothelioma, another lung cancer, though the data presented is less clear. The effect of glucocorticoids appears to be consistent with cellular senescence. Coupling this finding with the dependence of lung cancer cell lines on IGF-1R signaling for survival prompts the authors to classified the induced state as "dormant" rather than senescent or quiescent. The growth arrest is traced to upregulation of the cell-cycle inhibitor CDKN1C, and in particular to GR binding at a single locus (CERES). Finally, the openness of CERES is correlated with overall CDKN1C expression and recruitment of the SWI/SNF complex. There is a lot to like in this paper.

My major issues are that some assertions are either overstated or under substantiated, a lack of rationale for some methods of genomic analysis, and that the authors may be overlooking the implications of their findings. The last point is dealt with first in my comments below. The manuscript is also difficult to read in places where the data and methods used to make an assertion are not clear. These have been delineated in the minor comments.

Major Comments:

- 1) In my view the most impactful implication of the finding that glucocorticoids decrease growth, but enhance survivability of cancer cells is that they should probably not be used as an anti-nausea drug in lung cancer patients. As has been shown in the work of Suzanne Conzen and other in breast cancer, glucocorticoids can appear to help shrink the tumor by arresting growth, but because they also increase survival, in this case it appears by stimulating IGF-1R signaling, they work in opposition to chemotherapy. The authors do not explore whether "dormancy" confers protection against lung cancer chemotherapies. Instead they state in the abstract that the findings "illustrate the importance of GR signaling in solid cancer biology and set the stage for therapeutic application". I disagree with this statement. The importance of GR in solid tumors has been worked on by other groups, but I do not see how the study sets the stage for therapeutic application – it really cautions *against* therapeutic application.
- 2) The difference between "dormancy", senescence, and quiescence has not been made clear, though a lot of words are devoted to the terms. The authors need to clearly define the properties of cellular senescence and quiescence, then show how the state that glucocorticoids induce is different from those. I would argue that they should not only show some differences, but to be clear about how those differences warrant a new definition of cellular state. For example – how is dormancy the same and different from senescence? If it is that the cells not only stopped dividing, but have enhanced survival (through IGF-1R), is this really different from senescence, which may have different survival pathways turned on? Or is it a variant of senescence that does not need a new term, but by calling it something different confuses the field. If it is importantly different in ways that will affect how we understand the biology of non-dividing cells or treatment of cancer, these differences should be stated explicitly.
- 3) I disagree with the authors definition of direct regulation of genes by transcription factors. For FOXO1, IRS1, and CDKN1C the authors show gene regulation in response to glucocorticoids, proximal genomic binding, epigenetic marks (in the case of CDKN1C), and evidence of looping of these binding

regions to promoters. These data *provide evidence that the genes are likely* to be direct targets of GR though binding in these regions – but to state that GR, for instance “regulates CDKN1C expression through a novel distal enhancer” requires the authors to show that binding to this region is *required* for CDKN1C regulation, rather than that it is correlated. This issue could be ameliorated by buffering the language used. “Direct” control of genes is also stated on P4, line 91. What is the evidence in this case that the genes are directly regulated?

4) Why is high dose hydrocortisone used rather than dexamethasone, which is what is typically used in the clinic? Although hydrocortisone is the endogenous ligand, the levels used far exceed what would even occur physiologically and, adjusted for dose and half-life, might be less than pharmacological doses of dex. Therefore, because hydrocortisone is not commonly used in cancer treatment, the authors are studying neither endogenous nor relevant chemotherapeutic glucocorticoid signaling.

5) The gene expression analysis in the CDKN1C KO vs WT +/- GCs is curious. The authors have chosen to assert that a “GR activity signature” is unchanged – although there are some changes where the enrichment peaks. They also classify these as “direct targets” of GR – what is the evidence that they are direct targets? They have also chosen to break out other known sets, E2F and cell cycle genes, and measure differences by GSEA. This provides a visual difference, but where are the statistics? Why compare these gene sets by GSEA? If you are interested in identifying genes whose regulation by GCs has changed in CDKN1C KO versus wild type it is best to incorporate the interaction into the model in DESeq2 (genotype + treatment + genotype:treatment). Genes whose regulation (rather than just expression) has changed upon CDKN1C can then be gleaned from the interaction group.

6) The evidence that GC treatment “induces” contacts between CERES and the promoter of CDKN1C is overstated. There is clearly evidence of an association between these two loci in the absence of GCs – that association appears to be enhanced. It is not clear whether the enhancement, though visible in the plots shown, is significant.

7) You identify a correlation between GR levels, CERES accessibility, and CDKN1C expression in TCGA data sets. Presumably these are all in the absence of added GCs. Does this mean that CERES accessibility is associated with basal CDKN1C? How does this work with your model that GC-induced expression of CDKN1C is what causes senescence or dormancy? Shouldn't they already be senescent if CDKN1C is already elevated? This should be explained.

Minor:

1) In the abstract, the statement “the molecular mechanisms of glucocorticoid action in solid tumors remain elusive”. I have a couple of issues with this statement. First, the classification of solid tumors. Lymphomas are solid tumors for which glucocorticoids are a cornerstone of treatment and are very effective. Metastasizing cancers may or may not be considered solid, and we do not know the effect of glucocorticoids on these. Perhaps the classification should be “non-lymphoid solid cancers”. Or list the cancers you consider then define them as the non-lymphoid solid cancers. Second, many groups are studying the effect of glucocorticoids on breast and other reproductive cancers. These groups, including the Conzen and Lange groups, have done a great deal to characterize the effect of glucocorticoids on these cancers. I think the question you are really trying to answer is whether there is a property of solid cancers that ties them together as far as their response to glucocorticoids goes.

2) Also in the abstract, the effect of IGF-1R was identified and studied only in lung cancer models – it is not appropriate to lump it into having a role in all solid cancers.

3) Page 2, line 30. Glucocorticoids can regulate thousands of genes in some cells. Check out studies of leukemias.

4) Page 2, Line 38+. Glucocorticoids are not used for all liquid cancers – they are only effective in lymphoid lineage cells. So not AML or other myeloid lineage leukemias (See Markus München's work). The distinction you might want here is lymphoid versus non-lymphoid and solid.

5) Page 3, line 56. The wording of the last sentence of the intro is odd – to call regulation of CDKN1C “universal” but only in “multiple” tumors is odd. Perhaps maintaining an open CERES is commonly associated with high CDKN1C in solid tumors? Universal is a very strong classifier.

6) Page 3, line 61: The classification scheme for Figure 1a is not described well enough to understand the figure. Are they classified by gene mutation? Expression level? If either, how are mutants or expression levels judged.

7) P4, line 76, The authors do not show the data that “no signs of apoptosis were detected”.

8) P4, line 82. The p53 pathway is not activated, but is the expression level of p53 itself changed?

9) P4, Line 84. The authors state that ligand removal “reversed” growth inhibition. The data indicate that proliferation rate increases, but not to untreated rates. Perhaps “recovers to some extent”.

10) P4, line 87 and Figure S1D. I do not understand the different treatments and what they are supposed to show. This experiment requires further explanation.

11) I'm a little confused by the "GR activity" definition (P4, lines 88-90). How are the 253 genes chosen? I don't believe the TCGA data set has gene expression data for all of these cells +/- glucocorticoids, so are these genes picked from another data set? If so, and the TCGA is not +/- glucocorticoids, then the increased expression of these genes implies that they are regulated by GR in a ligand independent manner, or that the glucocorticoids present without being added (either in serum or in the body of tumors extracted from patients). I just need more explanation about how this gene set was identified and why it can be used as an indicator of GR "activity" in the absence of hormone. Unless I'm missing something.

12) Page 5, Lines 107 on. Synergistic drug interaction has a very specific definition that requires measurement at multiple concentrations for each drug and then calculating synergy by a variety of methods. A single drug concentration can enhance the efficacy or toxicity of another drug, but classification as synergistic requires further study.

13) Page 5, line 123. I find figures 2d and S2e confusing. The statement in the text is that treatment with GCs led to a decrease in Ki67 expression and Rb phosphorylation, and that Rb levels don't change. I find the figures confusing. If a P-Ab is used for Ki67, it looks like phosphorylation levels are down. This could be explained by a decrease in Ki67 expression. If this is not a P-Ab, then levels are down, but then why is there a No p-Ab row for Ki67? Similarly, for figure S2e, the text states that it measures Rb protein levels – is a No p-Ab antibody used for Rb when comparing +/- GCs? If so, why is there a No p-Ab row? The immunostaining is beautiful, but quantification is missing. Westerns might be a more quantitative way to measure changes in protein or phos-protein levels.

14) Page 6, paragraph beginning on line 132. The focus is on signaling downstream of IGF-1R, but did GCs cause a change in IGF-1R expression? Or phosphorylation? The manuscript states that GCs dephosphorylate insulin signaling pathways including IGF-1R, but Table S1 shows that's only true for H2122. I don't understand the argument that GC-induced phosphorylation of H2122 would make an inhibitor work less synergistically. So is the model that GCs do nothing to IGF-1R, but attenuate downstream signaling? I'm confused.

15) Page 7, Line 160. The finding that CDKN1C is upregulated by GCs in both lung cancer cells and mesothelioma is taken as evidence of "conservation of this route" beyond lung cancer. Conservation would mean that the gene is required for survival of the tumor – which would be overstated – I think at this point you mean "consistent between two cancers".

16) Page 7, Lines 164-168 describe that CDKN1C interacts with other CDKs, although the method for testing this is not stated in the text, only the legend of Figure 3, but how this experiment is done is not described. You are asking the reader to do too much work evaluate the evidence for a claim. Also, for Figure 3d there should be a negative control, as everything is enriched, and it could be that all proteins in the proteome are enriched.

17) Page 7, Line 168 – "instant" upregulation. The earliest timepoint is 1 hour – it is a stretch to call this instant.

18) Page 7, Line 177 – although the immunofluorescence images are beautiful, the level of knockout/knockdown is not quantified - and some CDKN1C is evident in H2122 KO cells after GC treatment.

19) Page 8, Line 189 – "In the absence of GCs" WT and KO H2122 cells "Have identical transcriptomes." The MA plot shows that they are not identical, with gene expression changing by up to 5 fold or more. The change might not be significant (although it is not stated whether or how this might have been tested), but there are differences. Also, in this MA plot there are curious horizontal stripes of genes that change the same amount – this suggests some bias in the RNA-seq, or that it is under-sequenced.

20) Page 9, Line 216: that p300 recruitment is "Exclusive to enhancer 1" is overstated. It is certainly most pronounced at enhancer 1, but there is some recruitment at Enhancer 2.

21) Page 10, Line 236: It is overstated to claim that CERES has been "fully characterized". This implies that all elements of the interaction have been systematically changed to understand function.

22) Page 10, Line 248: "Almost entirely absent" is overstated. The regulation of genes is attenuated in H1975 and H460 compared to the other cell lines.

23) Page 11, Line 254: NCOA1 and NRIP should be labeled in Figure 5d.

24) Page 11, Line 250: "GR itself was detected at similar levels" by RIME. If GR is being IP'd, would its level be expected to be similar? If there's less GR IP'd, then presumably all IP'd proteins would be down and levels would look similar after normalization.

25) Page 11, Lines 263-265. I do not understand what is meant by a "nominal" p value. Is this just a p value?

26) Page 12, Line 276. You observe GR and SWI/SNF at CERES in cervical cancer models – perhaps it is mentioned somewhere, but is this true of the other models (lung, mesothelioma) that you use? Why did you switch to a cervical cancer model at this point?

27) Page 18 – Paragraph starting on line 435. I don't understand the "link" that is being made between the effect of GCs in normal tissue and the effect on tumors. Also, on line 440 – is GR expression lower in all cancers compared to normal tissue? Some more detailed information would be useful.

Reviewer #2 (Remarks to the Author):

Prekovic-S,... ..Zwart-W, Glucocorticoid receptor triggers a novel mode of cell dormancy in solid cancers
Submitted to Nature Communications

In this manuscript, Zwart and colleagues employ gene regulatory analyses including Hi-C-, ChIP-seq and 4C-seq assays as well as mass spec IPs to characterize anti-proliferative glucocorticoid (GC) effects via IGF-1R signaling, and, more specifically, the CDKN1C locus (encoding p57KIP2), in five non-small cell lung cancer (NSCLC) cell lines. The authors found that binding of the glucocorticoid receptor (GR), upon recruitment of the histone acetyltransferase p300, to an enhancer element – named CERES (CDKN1C Enhancer Regulated by Steroids) – drives p57KIP2 expression via long-distance chromatin interactions. Based on pathway enrichment analyses, they concluded four complexes (nuclear transcription factor, SWI/SNF, mediator and RNA pol II) to participate in the active GR interactome. Furthermore, correlative analyses linked SWI/SNF defects in lung cancer patients to reduced GR activity. Genetic interference with eight SWI/SNF complex members demonstrated modulation of p57KIP2 expression. Analysis of ATAC data of a TCGA lung cancer cohort unveiled an association between high p57KIP2 expression and chromatin accessibility of the CERES element.

The paper is a technically and bioinformatically intense approach to pinpoint GC action via direct GR binding to a regulatory element as driver of cancer cell dormancy. While the first part of this goal is well performed – albeit loaded with (too) many global surveys such as GSEA, pharmacological screens, and bioinformatical analyses (like propensity scores) – classic molecular and genetic probing is less central (targeting, for instance, the SWI/SNF complex members, but not genomic elements, or the IGF-1R cascade, to study biological consequences in adequate tumor model systems).

Whether GC induces a lasting cell-cycle arrest via or beyond p57KIP2 through downstream signaling that does not involve direct genomic interaction of the GR at the CDKN1C locus, is not at all addressed and considered here.

While referring to a lung adenocarcinoma TCGA patient cohort and presenting interesting correlations between GR activity and genomic lesions in components of its interactome, very little translational insights are provided into p57KIP2-mediated dormancy, and its impact on long-term tumor fate in cancer patients. This is an almost exclusive (but dense and well performed) genomics paper, not a manuscript in which "GR-mediated CDKN1C regulation would have been truly established as a critical tumor suppressive axis in solid tumors", as claimed by the authors in the last sentence of the results section.

Major concerns and comments

1. Fig. 2a to visualize GC pre-treatment followed by vs. co-treatment with library compounds is not correct. The reader can only assume that the readout of this drug screen was survival – which is critical, since it is conceptually not obvious why the claimed GC-induced and IGF-1R-mediated cell dormancy should shift to cell death if IGF-1R signaling is blocked, or why acutely (by exogenous GC supply) enhanced IGF-1R signaling should create a vulnerability to IGF-1R inhibitors.

2. The authors focus on two genes claimed to mediate GR impact on the IGF-1R pathway: FoxO1 and IRS2. Functional genetics, i.e. knockdown/CRISPR experiments are required to demonstrate the actual dependency of the dormancy phenotype in response to GC in cell lines H1944, A549 and H2122.

3. While the manuscript addresses a putative enhancer element of p57KIP2 expression, investigation of GC-mediated signaling leading to transcriptional control of the CDKN1C promoter – although there is ample literature on its regulation – falls short. The point is not solely whether and where GR binds to regulatory elements within this locus region, but whether central downstream signaling may transactivate. GC induction of p57KIP2 expression is known for more than two decades (Samuelsson-MK et al., Mol Endocrinol 1999), and a GR-responsive element has been identified in the promoter of the CDKN1C gene in 2003 (Alheim-K et al., J Mol Endocrinol) – a finding questioned by the authors here – with numerous additional publications on this topic (e.g. Kaur-M et al., Mol Pharmacol 2008).

Certainly, in the year 2020 molecular analyses of regulatory elements are technically much more advanced and lead to much better resolution. However, the functional interpretation of GC action in NSCLC cells cannot be reduced to physical GR interaction with regulatory elements. Statements such as “...all suggest that Enhancer 1 is the main regulatory element through which GR regulates CDKN1C gene expression...” may be formally correct (with respect to GR binding), but do not reflect the complexity of a GR-governed signaling network driving p57KIP2 expression in a GC-dependent manner, as known for long.

4. In Fig. S1e, f the authors present a large number of human lung tumors analyzed regarding their GR activity based on the Z-score of 253 genes under direct GR control. To provide more biological evidence for the biomedical relevance of their findings, analyses of the GR complex from primary tumor material would be highly appreciated. And: what is the clinical distinction between GR-active vs. GR-inactive lung cancer patients in terms of treatment responsiveness and long-term outcome?

5. And: what about epidemiologic data on solid cancer incidence collected from long-term steroid-exposed patients, e.g. with autoimmune diseases or multiple myeloma?

Minor concerns

1. Definitive wording such as “Here, we explain the molecular mechanisms of glucocorticoid response in solid cancer models” in the abstract is a heavy overstatement and should be omitted. “Here, we elucidate molecular...” would sound more appropriate.

2. Experiments – in legends and the methods section – lack mandatory information; e.g. in Fig. 1b, c “cells treated with glucocorticoids (GC)” is simply insufficient: what steroid was used? At what dose? For how long?

3. Fig. 1c “no signs of apoptosis” – where is the data?

4. Fig. 1d: since H1975 and H460 did not respond to GC with a proliferative slow-down, was there also no marked effect in terms of protein dephosphorylation upon GC? And the obvious question whether all five cell lines exhibit comparable GC/GR translocation to the nucleus is only addressed in Fig. 5a.

5. The paper is quite difficult to read due to its nebulous information about actual experimental conditions. For instance, Fig. 3d, claimed in the main text to show interaction of p57KIP2 with certain cell-cycle regulators, presents R1-4 (abbreviations for “technical replicates 1-4?”, conducted in an unknown cell line) as “Z-scaled PSM scores” – which to at least partially understand forces the reader to jump to the materials section.

6. I’m not sure that Fig. 3j, k convincingly shows that “...changes in E2F target genes and other cell-cycle-related genes typically induced by GCs were entirely absent in the p57-KO model, indicating that p57 is the sole driver in the growth arrest phenotype”.

Reviewer #3 (Remarks to the Author):

Review Prekovic et al., Nature Communications 2020: "Glucocorticoid receptor triggers a novel mode of cell dormancy in solid cancers".

The manuscript describes how glucocorticoids that are used to treat the side-effects of chemotherapy, might in fact also influence the behavior of the cancer cells by inducing a dormancy-like state. Using a chemical-screening approach, they show that this state makes cancer cells more susceptible to IGF-pathway inhibitors. Accordingly, in a xenograft model, co-treatment with GCs and an IGF inhibitor showed synergistic inhibition of tumor growth. Analysis of RNA-seq data from several cell lines in which GR induces this dormancy-like state unveils CDKN1C as candidate gene responsible for this phenotype. Several omic approaches show that this gene is indeed regulated by GR, requires interaction with SWI-SNF components and is likely regulated by a specific enhancer that "loops" to the CDKN1C promoter in response to GC treatment. Using genome-editing, the authors further show that induction of the dormancy-like state in cell lines requires an intact CDKN1C gene. Finally, analysis of TCGA data, specifically data regarding the chromatin accessibility of the enhancer they identify, shows quite convincingly that openness of this enhancer and CDKN1C expression correlate.

Overall, I think this manuscript is quite interesting and provides convincing evidence that the GC-dependent regulation of CDKN1C is responsible for the GC-induced dormancy-like state. It uncovers interesting details regarding the regulation of this gene by GR and about co-treatment with GCs and IGF-inhibitors as a potential treatment for solid cancers. With some modifications I think it would be publishable.

Some remarks/Questions:

- What was the rationale for looking at the 5 cell lines you chose initially?
- Fig. 2a: I find this cartoon confusing. I think you either pre-treat or co-treat right? I am not sure if this cartoon explains the experimental set-up effectively....
- Related to this, how were synergistic effects defined in your study?
- I don't think you have shown that p57 is the sole driver of the growth arrest phenotype as claimed on page 8. To make this claim, I think you would need to show that simply overexpressing p57 is enough to induce this state. I think you have shown that it is necessary, but not if it is sufficient.
- Similarly, I do not agree that you have proven that the CERES enhancer is responsible for the regulation of the CDKN1C gene. I think your data (especially the TCGA data) provides very convincing indications that this is the case, but ultimate proof would require removing this enhancer to check if GR-dependent regulation still occurs in the absence of the CERES enhancer.
- Given that SWI-SNF components are cofactors for many TFs, it would be good to know if and how the knockdowns influence the basal expression level of the CDKN1C gene and not just the effect on the GC-induced level.
- The results of the TCGA analysis showing that the openness of the CERES enhancer correlates very strongly with expression of CDKN1C is really nice. Any SNPs linked to cancer susceptibility mapping to this locus?
- Maybe you could add some speculation to the discussion regarding the nature etc of this novel mode of dormancy you propose. How is it different from other types of dormancy and how might this be relevant?

Some typos:

- Line 65: Incorrect reference to figure?! I think this should be Figure S1C
- Line 222: referred should be referred.
- Line 326: Remove "been"

Reviewer #1 (Remarks to the Author):

The manuscript entitled “Glucocorticoid receptor triggers a novel mode of cell dormancy in solid cancers” explores the effect glucocorticoids on solid cancers, with a particularly deep analysis of non-small cell lung cancer cell lines. Glucocorticoids are a cornerstone in the treatment of lymphoid cancers but have been used in the regimen of prostate cancers, and as an anti-nausea drug to ameliorate the effects of chemotherapies in breast, lung, and other cancers. Their use in these “solid” tumors has been controversial, with some studies showing that glucocorticoids attenuate the effect of chemotherapy on breast cancer tissue, and can even substitute for androgens in castration resistant prostate cancer. A more thorough understanding of how glucocorticoids affect these solid tumor cells is important for understanding their use in the clinic. This study provides important data and an interesting model for how GCs may affect other cancers.

Overall, there is an impressive amount of data in this study on both the cellular phenomena and the mechanism behind them. The authors analyze the data in some useful ways as well. The authors clearly demonstrate that hydrocortisone (which is the same as the endogenous glucocorticoid cortisol) attenuates cell growth in the lung cancer lines without inducing cell death. Glucocorticoids may also have this effect mesothelioma, another lung cancer, though the data presented is less clear. The effect of glucocorticoids appears to be consistent with cellular senescence. Coupling this finding with the dependence of lung cancer cell lines on IGF-1R signaling for survival prompts the authors to classified the induced state as “dormant” rather than senescent or quiescent. The growth arrest is traced to upregulation of the cell-cycle inhibitor CDKN1C, and in particular to GR binding at a single locus (CERES). Finally, the openness of CERES is correlated with overall CDKN1C expression and recruitment of the SWI/SNF complex. There is a lot to like in this paper.

My major issues are that some assertions are either overstated or under substantiated, a lack of rationale for some methods of genomic analysis, and that the authors may be overlooking the implications of their findings. The last point is dealt with first in my comments below. The manuscript is also difficult to read in places where the data and methods used to make an assertion are not clear. These have been delineated in the minor comments.

We are delighted the reviewer appreciates our work, and finds ‘there is a lot to like in the paper’. We do however regret the reviewer found particular aspects overstated, under substantiated or difficult to read. We have edited the text, provided numerous experiments and additional computational analyses of both newly added and already existing datasets, as listed in detail in the point-by-point response below. We would like to express our gratitude to the reviewer for the insightful and detailed suggestions and feedback, which clearly helped to improve both the quality and readability of our work.

Major Comments:

1) In my view the most impactful implication of the finding that glucocorticoids decrease growth, but enhance survivability of cancer cells is that they should probably not be used as an anti-nausea drug in lung cancer patients. As has been shown in the work of Suzanne Conzen and other in breast cancer, glucocorticoids can appear to help shrink the tumor by arresting growth, but because they also increase survival, in this case it appears by stimulating IGF-1R signaling, they work in opposition to chemotherapy. The authors do not explore whether “dormancy” confers protection against lung cancer chemotherapies. Instead they state in the abstract that the findings “illustrate the importance of GR signaling in solid cancer biology and set the stage for therapeutic application”. I disagree with this statement. The importance of GR in solid tumors has been worked on by other groups, but I do not see how the study sets the stage for therapeutic application – it really cautions *against* therapeutic application.

We thank the reviewer for this thoughtful comment. In line with the reviewer’s comments we edited the manuscript and now additionally focus on the observation that glucocorticoids diminish efficacy of various compounds. Firstly, to further solidify these findings, we have performed an additional small size compound

screen (318 drugs) in H2122 cell line confirming our findings in H1944 cell line (**Figure 2b**). Furthermore, to learn more about the specific drug categories that glucocorticoids are inducing resistance to, we performed a CSgator compound set analysis (Park, Sera, et al. "CSgator: an integrated web platform for compound set analysis." *Journal of cheminformatics* 11.1 (2019): 17.). This unbiased analysis revealed that glucocorticoids significantly and selectively decreased efficacy of Threonine protease, kinase, guanylate cyclase, and structural protein inhibitors. Most, importantly among these drugs are the ones used for lung cancer treatment such as trametinib, methotrexate, dabrafenib, doxorubicin, and docetaxel, as presented in **Figure S3b**. In agreement with our findings and the reviewers comment we have also made changes to the abstract, highlighting the caution that should be considered in using GR agonists, as these may diminished applicability of various clinically applied therapeutic agents.

2) The difference between “dormancy”, senescence, and quiescence has not been made clear, though a lot of words are devoted to the terms. The authors need to clearly define the properties of cellular senescence and quiescence, then show how the state that glucocorticoids induce is different from those. I would argue that they should not only show some differences, but to be clear about how those differences warrant a new definition of cellular state. For example – how is dormancy the same and different from senescence? If it is that the cells not only stopped dividing, but have enhanced survival (through IGF-1R), is this really different from senescence, which may have different survival pathways turned on? Or is it a variant of senescence that does not need a new term, but by calling it something different confuses the field. If it is importantly different in ways that will affect how we understand the biology of non-dividing cells or treatment of cancer, these differences should be stated explicitly.

This an excellent point. As discussed in various reviews on the topic (e.g. Gorgoulis, Vassilis, et al. "Cellular senescence: defining a path forward." *Cell* 179.4 (2019): 813-827.; and Terzi, Menderes Yusuf, Muzeyyen Izmirli, and Bulent Gogebakan. "The cell fate: senescence or quiescence." *Molecular Biology Reports* 43.11 (2016): 1213-1220.), it became clear that classical definitions of senescence and quiescence only capture a part of the diversity in cell cycle arrest biology. While our results showed that our cells exposed particular features of senescence (but not all) and quiescence (but not all), we felt that either term would not capture the phenotype we observed. We have now amended the discussion to explicitly say which attributes of GR-induced cell cycle arrest have been associated to senescence and which to quiescence. As our findings represent a first report in which X-gal and senescence-based signatures are enriched but the phenotype is reversible and activation of growth signaling is observed, we believe this is not a variant of senescence nor quiescence. To address the reviewers' comment on the name of the growth arrest we have now changed the title, abstract, and removed the term we coined previously. The cellular state induced has now been described as a “reversible drug-tolerant dormancy state”.

3) I disagree with the authors definition of direct regulation of genes by transcription factors. For FOXO1, IRS1, and CDKN1C the authors show gene regulation in response to glucocorticoids, proximal genomic binding, epigenetic marks (in the case of CDKN1C), and evidence of looping of these binding regions to promoters. These data *provide evidence that the genes are likely* to be direct targets of GR though binding in these regions – but to state that GR, for instance “regulates CDKN1C expression through a novel distal enhancer” requires the authors to show that binding to this region is *required* for CDKN1C regulation, rather than that it is correlated. This issue could be ameliorated by buffering the language used. “Direct” control of genes is also stated on P4, line 91. What is the evidence in this case that the genes are directly regulated?

We would like to thank the reviewer for highlighting this critical issue, and we fully agree with the reviewer that the evidence that was provided in the previous version of our manuscript, was insufficient to justifiably support the claim of direct regulation of target genes. To address this, we have now performed CRISPR-based enhancer perturbations to further investigate the mode of regulation of *CDKN1C*. Specifically, CRISPR-Cas9 was used to cut each of the three potential enhancer sites out of the genome, in which cutting of *CDKN1C* gene and *ABCB1* promoter were used as positive and negative controls, respectively. The new data now shows that physical removal of enhancer 1 (CERES enhancer) significantly decreased GR-driven induction of *CDKN1C* expression

in a polyclonal cell population, which is accompanied by a rescue of the cells from glucocorticoid receptor induced growth arrest. Importantly, we show that cutting of enhancers 2 and 3 does not lead to altered induction of growth arrest and *CDKN1C* induction. These findings can be appreciated in **Figures 4e and f**. As for FOXO1/IRS1 -which we feel were of added value, but not critical for our message- we have now moved the panels containing these data to the supplementary files and have amended the text according to the suggestions, stating that expression of these genes correlated with GR binding without making causal statements.

4) Why is high dose hydrocortisone used rather than dexamethasone, which is what is typically used in the clinic? Although hydrocortisone is the endogenous ligand, the levels used far exceed what would even occur physiologically and, adjusted for dose and half-life, might be less than pharmacological doses of dex. Therefore, because hydrocortisone is not commonly used in cancer treatment, the authors are studying neither endogenous nor relevant chemotherapeutic glucocorticoid signaling.

We thank the reviewer for this comment. While most data streams were indeed generated from cells treated with hydrocortisone at saturating conditions, we have also used dexamethasone in various other experiments throughout the manuscript. In addition, *in vitro*, unpublished work from the laboratory of Sebastiaan Meijsing (personal communication) shown that at saturating levels dexamethasone and hydrocortisone produce the same effects on RNA sequencing, as well as STAR sequencing level. In the new version of the manuscript we also now include new data produced on primary mesothelioma cell lines treated with various concentrations of dexamethasone and hydrocortisone showing the same effects (**Figure S2**). Furthermore, we have treated mouse-xenograft models with A549 cells engrafted with dexamethasone (**Figure S4a and b**), showing that this indeed blocks the growth of these tumors, and induces vulnerability to IGFR inhibitors *in vivo*. In addition, we have now performed RNA sequencing of H1944 xenograft tumors treated with a clinically-relevant dose of dexamethasone showing that this indeed leads to upregulation of classical GR target genes (e.g. FKBP5 and PER1), as well upregulation of *CDKN1C* expression *in vivo* (**Figure S3f-h**). We have now included a table containing all the required information on steroid used, concentration and duration, for each panel found in the manuscript (**Table S6**).

5) The gene expression analysis in the *CDKN1C* KO vs WT +/- GCs is curious. The authors have chosen to assert that a "GR activity signature" is unchanged – although there are some changes where the enrichment peaks. They also classify these as "direct targets" of GR – what is the evidence that they are direct targets? They have also chosen to break out other known sets, E2F and cell cycle genes, and measure differences by GSEA. This provides a visual difference, but where are the statistics? Why compare these gene sets by GSEA? If you are interested in identifying genes whose regulation by GCs has changed in *CDKN1C* KO versus wild type it is best to incorporate the interaction into the model in DESeq2 (genotype + treatment + genotype:treatment). Genes whose regulation (rather than just expression) has changed upon *CDKN1C* can then be gleaned from the interaction group.

We thank the reviewer for this comment, we have now performed the analysis accordingly. Firstly, we have included the statistics to substantiate the conclusions statistically (**Figure 3i-k**). Secondly, we have performed additional analysis using DESeq2, as suggested by the reviewer, and now include a volcano plot, heatmap, and a gene-list analysis (*Raudvere, Uku, et al. "g: Profiler: a web server for functional enrichment analysis and conversions of gene lists (2019 update)." Nucleic acids research 47.W1 (2019): W191-W198.*) of all the genes changed between p57-WT and p57-KO in GC treatment condition. The p57-KO cells treated with glucocorticoids do not exhibit reduction in expression of genes having a cell cycle-related role, as can be appreciated in **Figure 5Sg-i**, thus confirming the findings in the main figure.

In respect to "direct" GR target genes we have used an activity signature generated in our laboratory to test this, this signature is based on RNA-sequencing and ChIP-sequencing *in vitro* experiments with cell treated with vehicle or glucocorticoids. We have now changed this in the text, referring to these as "active GR-associated genes", not "direct target genes". The conclusions based on gset enrichment analysis found in the manuscript remain the same and are now further strengthened as no GR-target genes (e.g. *FKBP5*, *PER1*, *DUSP1*) were

identified to be different between glucocorticoid treated WT and p57-KO cells, as seen in the volcano plot (**Figure S5g**). In fact, the only gene that was found to be statistically significantly higher expressed in the WT cell line in comparison to p57-KO cell line was p57 which target gene for our CRSIPR-Cas9 (**Figure S5g**).

6) The evidence that GC treatment “induces” contacts between CERES and the promoter of CDKN1C is overstated. There is clearly evidence of an association between these two loci in the absence of GCs – that association appears to be enhanced. It is not clear whether the enhancement, though visible in the plots shown, is significant.

We thank the reviewer for raising this point, as indeed the contacts are pre-existing and just enhanced by glucocorticoid treatment, in line with previous publications on the topic (*D'Ippolito, Anthony M., et al. "Pre-established chromatin interactions mediate the genomic response to glucocorticoids." Cell systems 7.2 (2018): 146-160.*). This has now been corrected textually – indicating that the contacts are enhanced, not induced. Furthermore, we have now included a statistical analysis (Wilcoxon test) of the data supporting our conclusions of interaction stabilization following GR activation. This can be seen in **Figure 4c and d**. In addition, we also provide now 4C analysis of two cell lines that do not respond to GCs, that is no increase in *CDKN1C* expression was found. In agreement with these findings, stabilization of the contact between of CERES and *CDKN1C* promoter was not found upon GR activation. These data can be seen in **Figure S6b**.

7) You identify a correlation between GR levels, CERES accessibility, and CDKN1C expression in TCGA data sets. Presumably these are all in the absence of added GCs. Does this mean that CERES accessibility is associated with basal CDKN1C? How does this work with your model that GC-induced expression of CDKN1C is what causes senescence or dormancy? Shouldn't they already be senescent if CDKN1C is already elevated? This should be explained.

The data we present in Figure 6 firstly suggests that accessibility of CERES enhancer is linked to expression of *CDKN1C*. Importantly, we show that this link is indeed related to glucocorticoid receptor expression, as correlations of CERES accessibility and *CDKN1C* are strengthened upon removal of tumor samples from the analysis, with low levels of GR expression. Furthermore, in **Figure S1i and j**, we show that indeed, senescence and growth arrest at least on level of gene expression is associated with activity of GR in these same tumors. This is in line with the notion that glucocorticoids from the circulation can affect intratumoral GR activity, therefore cancers would be affected even without the use of prednisone/dexamethasone. However, as many older patients do receive glucocorticoids for other ailments and this is not often well annotated we cannot exclude that our observations are also strengthened by this. The revised version of the manuscript contains additional information and a more in-depth discussion on this.

Minor:

1) In the abstract, the statement “the molecular mechanisms of glucocorticoid action in solid tumors remain elusive”. I have a couple of issues with this statement. First, the classification of solid tumors. Lymphomas are solid tumors for which glucocorticoids are a cornerstone of treatment and are very effective. Metastasizing cancers may or may not be considered solid, and we do not know the effect of glucocorticoids on these. Perhaps the classification should be “non-lymphoid solid cancers”. Or list the cancers you consider then define them as the non-lymphoid solid cancers. Second, many groups are studying the effect of glucocorticoids on breast and other reproductive cancers. These groups, including the Conzen and Lange groups, have done a great deal to characterize the effect of glucocorticoids on these cancers. I think the question you are really trying to answer is whether there is a property of solid cancers that ties them together as far as their response to glucocorticoids goes.

Following the reviewer's recommendation, we have now rephrased the abstract and have changed 'solid cancers' to "non-lymphoid solid cancers". As for previous studies by various groups, including the Conzen and Lange groups, we acknowledge the previous work in our discussion section and acknowledge that they have

done a lot of work to define the effects of glucocorticoids on cancer biology. In our paper we build upon this work and now focus on the direct effectors that mediate these phenotypes and use various techniques learn more about the growth arrest phenotype. We have now amended the abstract accordingly, indicating we aimed to identify molecular underpinnings and direct effectors of GR signaling in nonhematological cancers, focusing on lung cancer.

2) Also in the abstract, the effect of IGF-1R was identified and studied only in lung cancer models – it is not appropriate to lump it into having a role in all solid cancers.

We have now corrected this, and throughout the abstract indicate that our work mostly focuses on lung cancer.

3) Page 2, line 30. Glucocorticoids can regulate thousands of genes in some cells. Check out studies of leukemias.

As context impacts GR action between different cell types, and number of differentially expressed genes after a specific stimulus depends on duration, statistical cut-offs used, etc., this prevents us to make generalizable statement, we have now amended this to read:

“ultimately modulating expression of **a large number** of genes, across diverse cell types”.

4) Page 2, Line 38+. Glucocorticoids are not used for all liquid cancers – they are only effective in lymphoid lineage cells. So not AML or other myeloid lineage leukemias (See Markus München’s work). The distinction you might want here is lymphoid versus non-lymphoid and solid.

We thank the reviewer for pointing this error, which is now corrected in the revised version of the manuscript:

“While pharmacological agonists of the GR (e.g. Prednisone, and Dexamethasone) have been intensively used as therapeutics in treatment of **hematologic cancers**, for **non-lymphoid solid** (i.e. non-hematologic) cancers patients they are utilised solely as adjuvant treatment to alleviate symptoms caused by anti-cancer therapy.”

5) Page 3, line 56. The wording of the last sentence of the into is odd – to call regulation of CDKN1C “universal” but only in “multiple” tumors is odd. Perhaps maintaining an open CERES is commonly associated with high CDKN1C in solid tumors? Universal is a very strong classifier.

We agree with the reviewer, and reformulated accordingly by removing the word universal. Now the updated phrasing reads:

“Finally, using transcriptomics and chromatin accessibility data of clinical samples, we show that this mode of regulation **occurs** in multiple human **non-lymphoid solid cancer** types.”

6) Page 3, line 61: The classification scheme for Figure 1a is not described well enough to understand the figure. Are they classified by gene mutation? Expression level? If either, how are mutants or expression levels judged.

Indeed, the scheme of Figure 1a was not clear enough to communicate the message in its current form. Now, we have moved this figure to the supplementary materials (Figure S1a) and provided an additional explanation in the legend ensuring sufficient level of description. As stated, in this panel we depicted the mutation status (either wt or aberrant) of key lung cancer drivers as reported by the ATCC.

7) P4, line 76, The authors do not show the data that “no signs of apoptosis were detected”.

This statement was based on visual inspection of microscopy images, in which no apoptotic cells were detected. To provide unbiased direct assessment of apoptosis, we now performed PARP western blot for untreated and glucocorticoid treated cells. As can be appreciated in **Figure S1c**, GR activation did not induce PARP cleavage, indicating that apoptosis is not occurring. Furthermore, we have used an ANEXIN-V FACS based assay to show that GCs do not induce apoptosis in mesothelioma models (**Figure S2c**). In addition, we have used IHC to detect cleaved Caspase3 in the tumor samples of xenograft H1944 cells, and no cleaved Caspase 3 signal could be detected (**Figure S3i**). Based on this, we conclude that GR activation in our cell line models does not induce apoptosis.

8) P4, line 82. The p53 pathway is not activated, but is the expression level of p53 itself changed?

In order to address this, we have now performed p53 western blot for untreated and glucocorticoid treated cells. As can be appreciated in **Figure S1c**, we do not observe a change in p53 levels upon activation of GR across the five cell lines used.

9) P4, Line 84. The authors state that ligand removal “reversed” growth inhibition. The data indicate that proliferation rate increases, but not to untreated rates. Perhaps “recovers to some extent”.

We thank the reviewer as the assay initially performed to show this indeed did not illustrate the extent of the rescue. To address this point, we have now performed a more adequate assay, that is we have now included a real-time confluency analysis following pretreatment with vehicle or glucocorticoids. This indicates that the upon ligand withdrawal the growth-inhibition was lost and cells restarted proliferating to a comparable rate as the ones not exposed to glucocorticoids. This can be found in **Figure S1f**.

10) P4, line 87 and Figure S1D. I do not understand the different treatments and what they are supposed to show. This experiment requires further explanation.

We have now added the methods section and provided an additional reference which provides additional details.

“For OCR measurements the following reagents that **selectively inhibit mitochondrial function**⁹⁵ were added: oligomycin (1 μ M; **a ATP synthase (complex V) inhibitor**), FCCP (0,4 μ M; **an uncoupling agent that collapses the proton gradient and disrupts the mitochondrial membrane potential**), and rotenone (1 μ M; **a mitochondrial complex I inhibitor**) and antimycin A (1 μ M; **a mitochondrial complex III inhibitor**). Results were normalized to DNA content using nanodrop quantification (Thermo Fisher Scientific).”

11) I'm a little confused by the “GR activity” definition (P4, lines 88-90). How are the 253 genes chosen? I don't believe the TCGA data set has gene expression data for all of these cells +/- glucocorticoids, so are these genes picked from another data set? If so, and the TCGA is not +/- glucocorticoids, then the increased expression of these genes implies that they are regulated by GR in a ligand independent manner, or that the glucocorticoids present without being added (either in serum or in the body of tumors extracted from patients). I just need more explanation about how this gene set was identified and why it can be used as an indicator of GR “activity” in the absence of hormone. Unless I'm missing something.

The GR activity signature has been developed in our laboratory and is based on various RNA sequencing and GR ChIP sequencing experiments of numerous *in vitro* models treated or untreated with glucocorticoids. We have used these data and computationally optimized the signature to specifically predict of the activity of GR at the target gene level. The design of the GR signature and the pan-cancer application thereof is part of a separate manuscript that we are finalizing at the moment, and would therefore deviate too much from the main message as presented here to implement in full, into the current manuscript.

12) Page 5, Lines 107 on. Synergistic drug interaction has a very specific definition that requires measurement at multiple concentrations for each drug and then calculating synergy by a variety of methods. A single drug concentration can enhance the efficacy or toxicity of another drug, but classification as synergistic requires further study.

We completely agree with the reviewer, and have removed the synergism statement, as indeed studying synergy requires additional in-depth analyses that were not performed here. Instead we now indicate that the effect was enhanced by the dual treatment; a statement that we can justifiably make.

13) Page 5, line 123. I find figures 2d and S2e confusing. The statement in the text is that treatment with GCs led to a decrease in Ki67 expression and Rb phosphorylation, and that Rb levels don't change. I find the figures confusing. If a P-Ab is used for Ki67, it looks like phosphorylation levels are down. This could be explained by a decrease in Ki67 expression. If this is not a P-Ab, then levels are down, but then why is there a No p-Ab row for Ki67? Similarly, for figure S2e, the text states that it measures Rb protein levels – is a No p-Ab antibody used for Rb when comparing +/- GCs? If so, why is there a No p-Ab row? The immunostaining is beautiful, but quantification is missing. Westerns might be a more quantitative way to measure changes in protein or phospho-protein levels.

We apologize for this confusion, in the legends it was stated that 'No p-Ab' refers to 'no primary antibody' as described in the legends. However, we do see why this is confusing. Therefore, we have amended this and it now read "No primary Ab". In addition, we use RNA sequencing of the same xenografts to show that indeed the E2F network is down, something that concomitantly occurs after loss of Rb phosphorylation. These data can be found in **Figure S3h**.

14) Page 6, paragraph beginning on line 132. The focus is on signaling downstream of IGF-1R, but did GCs cause a change in IGF-1R expression? Or phosphorylation? The manuscript states that GCs dephosphorylate insulin signaling pathways including IGF-1R, but Table S1 shows that's only true for H2122. I don't understand the argument that GC-induced phosphorylation of H2122 would make an inhibitor work less synergistically. So is the model that GCs do nothing to IGF-1R, but attenuate downstream signaling? I'm confused.

We apologize for the confusion. We show that glucocorticoids lead to increase in phosphorylation of IGF-1R *in vitro* and *in vivo* using xenograft models. The manuscript does not state that GCs dephosphorylate insulin signaling, but instead lead to activation of this pathway. In H2122, we have shown using DepMap data that these cells are already dependent on IGFR signaling, as deletion of this gene leads to cell death; hence, no enhanced effect of treatment is to be expected in this setting. We have now added an additional *in vivo* xenograft model (A549 cells) to show that in addition to H1944, glucocorticoids induce vulnerabilities to two different IGFR inhibitors (Linsitinib and GSK-1838705A). This can be seen in **Figure S4a and b**.

15) Page 7, Line 160. The finding that CDKN1C is upregulated by GCs in both lung cancer cells and mesothelioma is taken as evidence of "conservation of this route" beyond lung cancer. Conservation would mean that the gene is required for survival of the tumor – which would be overstated – I think at this point you mean "consistent between two cancers".

We thank the reviewer for pointing this out. In line with the reviewer's suggestion, we have now removed this part of the sentence, as we agree it was flawed:

"This gene was also found upregulated in the H2795 mesothelioma cell line which was growth-arrested by GCs, but not in two GC-resistant mesothelioma models (Fig. S5a)."

16) Page 7, Lines 164-168 describe that CDKN1C interacts with other CDKs, although the method for testing this is not stated in the text, only the legend of Figure 3, but how this experiment is done is not described. You

are asking the reader to do too much work evaluate the evidence for a claim. Also, for Figure 3d there should be a negative control, as everything is enriched, and it could be that all proteins in the proteome are enriched.

We apologize for the omission. We have now included the statistical comparison of p57-IP and corresponding IgG-IP, that shows various cell cycle related proteins to be enriched in our bait-IPs (**Figure 3d**). In addition, these analyses show specific enrichment of these proteins, and that not the whole proteome was enriched in our analysis. The analysis was done on data produced on four independent, biological replicates in H2122 cell line, using IgG as negative control (also in four independent biological replicates; the data has been deposited to PRIDE). We have now included additional descriptions in the result section, and legends.

17) Page 7, Line 168 – “instant” upregulation. The earliest timepoint is 1 hour – it is a stretch to call this instant.

We have now reformulated this:

“Importantly, upregulation of CDKN1C mRNA (Fig. 3e) preceded the transcriptional downregulation of various cell cycle genes (including CCND3, and CCNE2) which was observed after four hours of GC treatment (Fig. 3f).”

18) Page 7, Line 177 – although the immunofluorescence images are beautiful, the level of knockout/knockdown is not quantified - and some CDKN1C is evident in H2122 KO cells after GC treatment.

As suggested we have now included the quantification of the p57 immunopositivity in the knockouts, preformed in three independent biological replicates. These quantifications indicate that expression was highly decreased in the KO cells, at statistically significant level. These new analyses can be found in **Figure S5c**.

19) Page 8, Line 189 – “In the absence of GCs” WT and KO H2122 cells “Have identical transcriptomes.” The MA plot shows that they are not identical, with gene expression changing by up to 5 fold or more. The change might not be significant (although it is not stated whether or how this might have been tested), but there are differences. Also, in this MA plot there are curious horizontal stripes of genes that change the same amount – this suggests some bias in the RNA-seq, or that it is under-sequenced.

To address this comment, we have now performed statistical analysis to show that knocking out p57 has no statistically-significant effect on basal gene expression, when compared to the cells with a non-targeting guide (**Figure S5e**). According to standard statistical parameters based on multiple-testing corrected p-value (recommended by Encode for RNA sequencing experiments) we detect 0 genes differentially expressed. While log-fold differences as can be observed in **Figure S5e** these are not statistically significance and are most likely the consequence of stochastic behavior of gene expression, observed especially for genes expressed at low levels as seen in **Figure S5f**. Furthermore, in terms of sequencing depth we have, as for all the experiments followed the Encode recommendations, and the horizontal-strips observed on MA-plots are typically seen for deferential expression experiments, especially for genes with low expression levels. In accordance with the reviewers’ comment and the newly provided data analysis have now rephrased our conclusions to:

“In the absence of GCs, no statistically significant differences in mRNA expression were detected between the p57-WT and p57-KO H2122 (Fig. S5e, f).”

20) Page 9, Line 216: that p300 recruitment is “Exclusive to enhancer 1” is overstated. It is certainly most pronounced at enhancer 1, but there is some recruitment at Enhancer 2.

Following the reviewers’ advice, we have now reformulated this section. The text was amended to the following:

“The active enhancers-associated factors²⁹, histone acetyltransferase p300 and H3K27Ac chromatin mark, were most pronounced at Enhancer 1 (Fig. 4b).”

21) Page 10, Line 236: It is overstated to claim that CERES has been “fully characterized”. This implies that all elements of the interaction have been systematically changed to understand function.

Following the reviewer’s recommendation, we have removed “fully characterized” from this sentence. The corrections are as follows:

“Collectively, we have discovered a GR driven enhancer that regulates CDKN1C gene through long-distance chromatin interactions, thereby controlling cell dormancy entry.”

22) Page 10, Line 248: “Almost entirely absent” is overstated. The regulation of genes is attenuated in H1975 and H460 compared to the other cell lines.

We agree with the reviewer, and revised the text accordingly:

“In contrast to that, GR-driven gene expression changes were observed in the GC-growth arrested cell lines (A549, H2122, and H1944), while this was **strongly attenuated** in the GC-unresponsive cell lines H1975 and H460 (Fig. 5c).”

23) Page 11, Line 254: NCOA1 and NRIP should be labeled in Figure 5d.

This has now been included in the new version of **Figure 5D** of the manuscript.

24) Page 11, Line 250: “GR itself was detected at similar levels” by RIME. If GR is being IP’d, would its level be expected to be similar? If there’s less GR IP’d, then presumably all IP’d proteins would be down and levels would look similar after normalization.

This could be the case, however this is used as a control to show that our comparisons were not influenced the variability in GR-IP levels. In the normalization the samples are all mean normalized and not normalized to the IP-bait.

25) Page 11, Lines 263-265. I do not understand what is meant by a “nominal” p value. Is this just a p value?

The nominal p-value estimates the statistical significance of the enrichment score for a single gene set. However, when you are evaluating multiple gene sets, you must correct for gene set size and multiple hypothesis testing. Because the p value is not adjusted for either, it is of limited value when comparing gene sets. The Gene Set Enrichment Analysis PNAS paper (Subramanian, Aravind, et al. "Gene set enrichment analysis: a knowledge-based approach for interpreting genome-wide expression profiles." Proceedings of the National Academy of Sciences 102.43 (2005): 15545-15550.) describes the p value statistic in the section titled Appendix: Mathematical Description of Methods. We refer to this paper in the methods section.

26) Page 12, Line 276. You observe GR and SWI/SNF at CERES in cervical cancer models – perhaps it is mentioned somewhere, but is this true of the other models (lung, mesothelioma) that you use? Why did you switch to a cervical cancer model at this point?

To show this, we used publicly available data sets, which unfortunately do not exist for lung and mesothelioma, but were available for cervical cancer models. We now mention this in the results section.

27) Page 18 – Paragraph starting on line 435. I don’t understand the “link” that is being made between the effect

of GCs in normal tissue and the effect on tumors. Also, on line 440 – is GR expression lower in all cancers compared to normal tissue? Some more detailed information would be useful.

We have amended this section providing additional details. We sincerely hope this reformulation increased clarity, and the additional information addresses the reviewer's concern.

“We speculate that the role of GR to induce cellular dormancy in cancer may be reminiscent of its signaling in normal tissue. In agreement with this, activation of GR has previously been linked to cell differentiation and lineage selection^{71,72}. As tumours are exposed to GCs **produced by the adrenal gland and released into the** circulation, this dormant state might be a feature of various early-stage human tumours, supported by the observation that GR expression is lower in **various cancer types (including lung, breast and prostate) as compared to normal tissue**, and that it may serve as a tumour suppressor ¹³. Furthermore, cell dormancy has been shown to be under circadian control in various tissues and stem cells^{73,74}. In relation to that, the well-known day-night rhythmic behavior of GC levels may also suggest that cell dormancy as well as **chemotherapy response** in cancer is subjected to circadian rhythm through **intratumoral** GR activity.”

Reviewer #2 (Remarks to the Author):

Prekovic-S,... ..Zwart-W, Glucocorticoid receptor triggers a novel mode of cell dormancy in solid cancers

Submitted to Nature Communications

In this manuscript, Zwart and colleagues employ gene regulatory analyses including Hi-C-, ChIP-seq and 4C-seq assays as well as mass spec IPs to characterize anti-proliferative glucocorticoid (GC) effects via IGF-1R signaling, and, more specifically, the CDKN1C locus (encoding p57KIP2), in five non-small cell lung cancer (NSCLC) cell lines. The authors found that binding of the glucocorticoid receptor (GR), upon recruitment of the histone acetyltransferase p300, to an enhancer element – named CERES (CDKN1C Enhancer Regulated by Steroids) – drives p57KIP2 expression via long-distance chromatin interactions. Based on pathway enrichment analyses, they concluded four complexes (nuclear transcription factor, SWI/SNF, mediator and RNA pol II) to participate in the active GR interactome. Furthermore, correlative analyses linked SWI/SNF defects in lung cancer patients to reduced GR activity. Genetic interference with eight SWI/SNF complex members demonstrated modulation of p57KIP2 expression. Analysis of ATAC data of a TCGA lung cancer cohort unveiled an association between high p57KIP2 expression and chromatin accessibility of the CERES element.

The paper is a technically and bioinformatically intense approach to pinpoint GC action via direct GR binding to a regulatory element as driver of cancer cell dormancy. While the first part of this goal is well performed – albeit loaded with (too) many global surveys such as GSEA, pharmacological screens, and bioinformatical analyses (like propensity scores) – classic molecular and genetic probing is less central (targeting, for instance, the SWI/SNF complex members, but not genomic elements, or the IGF-1R cascade, to study biological consequences in adequate tumor model systems).

Whether GC induces a lasting cell-cycle arrest via or beyond p57KIP2 through downstream signaling that does not involve direct genomic interaction of the GR at the CDKN1C locus, is not at all addressed and considered here.

While referring to a lung adenocarcinoma TCGA patient cohort and presenting interesting correlations between GR activity and genomic lesions in components of its interactome, very little translational insights are provided into p57KIP2-mediated dormancy, and its impact on long-term tumor fate in cancer patients. This is an almost exclusive (but dense and well performed) genomics paper, not a manuscript in which “GR-mediated CDKN1C regulation would have been truly established as a critical tumor suppressive axis in solid tumors”, as claimed by the authors in the last sentence of the results section.

We thank the reviewer for the insightful and detailed suggestions that clearly helped us to further improve the quality of our work. As can be appreciated in the point-by-point response to the reviewer comments below, we attempted to address the comments to the best of our capacities, and hope the reviewer is satisfied with the answered provided and finds the manuscript sufficiently improved for acceptance. In agreement with the reviewers comment throughout the manuscript we use several genetic models and drug inhibition to validate and explore the biological meaning of our findings, and provide an additional analysis of data in our manuscript to support our findings.

Major concerns and comments

1. Fig. 2a to visualize GC pre-treatment followed by vs. co-treatment with library compounds is not correct. The reader can only assume that the readout of this drug screen was survival – which is critical, since it is conceptually not obvious why the claimed GC-induced and IGF-1R-mediated cell dormancy should shift to cell death if IGF-1R signaling is blocked, or why acutely (by exogenous GC supply) enhanced IGF-1R signaling should create a vulnerability to IGF-1R inhibitors.

We would like to thank the reviewer for highlighting this unclarity. We have now removed the panel in question, as it had errors that made the interpretation of the data difficult. We now replaced the panel with another one depicting the design in greater details (**Figure S3a**) and have included a legend that extensively explains how the drug screen was performed. In addition, we have also in details explained this in the result section. Furthermore, we have validated the findings that long-term glucocorticoid treatment induces vulnerabilities to IGF-1R inhibitors in another lung cancer xenograft models (A549; **Figure S4a and b**).

2. The authors focus on two genes claimed to mediate GR impact on the IGF-1R pathway: FoxO1 and IRS2. Functional genetics, i.e. knockdown/CRISPR experiments are required to demonstrate the actual dependency of the dormancy phenotype in response to GC in cell lines H1944, A549 and H2122.

We thank the reviewer for raising this relevant issue. We have now rephrased the sentences in which we make these statements, as we merely wanted to propose that these genes may be relevant in induction of IGF-1R signaling pathway by glucocorticoids. Considering the comments of all the reviewers we moved the data on FOXO1 and IRS2 in the supplementary figures, as this was not the key finding we wanted to highlight in our manuscript, which is mostly focusing on *CDKN1C*. However, in order to address the reviewers' comment, we have now performed additional experiments using mass spectrometry in which we can show that inhibiting FOXO1 or IRS2 has a negative impact on glucocorticoid-induced changes in phosphorylation of various proteins of the IGF1R pathway (performed in three biological replicates). As these data do not significantly contribute to the general message we want to convey, potentially distracting from the main message, we would like not to include these data in the final version of the manuscript, but still provide this information for the reviewer's consideration.

Figure 1. Glucocorticoid treatment of H1944 cell line led to changes in phosphorylation status of several key genes of the IGF-1R pathway. These effects of glucocorticoids were diminished upon addition of FOXO1 and IRS2 inhibitors. The experiment was done in three biological replicates.

3. While the manuscript addresses a putative enhancer element of p57KIP2 expression, investigation of GC-mediated signaling leading to transcriptional control of the CDKN1C promoter – although there is ample literature on its regulation – falls short. The point is not solely whether and where GR binds to regulatory elements within this locus region, but whether central downstream signaling may transactivate. GC induction of p57KIP2 expression is known for more than two decades (Samuelsson-MK et al., Mol Endocrinol 1999), and a GR-responsive element has been identified in the promoter of the CDKN1C gene in 2003 (Alheim-K et al., J Mol Endocrinol) – a finding questioned by the authors here – with numerous additional publications on this topic (e.g. Kaur-M et al., Mol Pharmacol 2008).

Certainly, in the year 2020 molecular analyses of regulatory elements are technically much more advanced and lead to much better resolution. However, the functional interpretation of GC action in NSCLC cells cannot be reduced to physical GR interaction with regulatory elements. Statements such as “...all suggest that Enhancer 1 is the main regulatory element through which GR regulates CDKN1C gene expression...” may be formally correct (with respect to GR binding), but do not reflect the complexity of a GR-governed signaling network driving p57KIP2 expression in a GC-dependent manner, as known for long.

We thank the reviewer for this comment. We acknowledge and cite the previous papers that have been published on topic of glucocorticoid regulation of *CDKN1C* in the previous and current version of the manuscript. Most of the previous work has been done without the current genetic tools available at that time. The finding of the GR-responsive element in the promoter of *CDKN1C* was based on EMSA experiments, and in our paper, we perform GR ChIP sequencing in which we could not identify any binding of GR to the promoter. These findings do not dispute the prior ones and can probably be explained by the chromatin-context that is absent in EMSA experiments. We now included a statement on this possible explanation in the results section. To address the relevance of the reported enhancers on p57 regulation by glucocorticoids, we have now performed CRISPR-based enhancer perturbations to further investigate the mode of regulation of *CDKN1C*. In short, CRISPR-Cas9 was used to cut each of the three potential enhancer sites out of the genome, in which cutting of *CDKN1C* gene and *ABCB1* promoter were used as positive and negative controls, respectively. The new data now shows that physical removal of enhancer 1 (CERES enhancer) significantly decreased GR-driven induction of *CDKN1C* expression in a polyclonal cell population, which is accompanied by a rescue of the cells from glucocorticoid receptor induced growth arrest. Importantly, we show that cutting of enhancer 2 and 3 do not lead to altered induction of growth arrest and *CDKN1C* induction. These findings can be appreciated in **Figures 4e and f**.

4. In Fig. S1e, f the authors present a large number of human lung tumors analyzed regarding their GR activity based on the Z-score of 253 genes under direct GR control. To provide more biological evidence for the biomedical relevance of their findings, analyses of the GR complex from primary tumor material would be highly appreciated. And: what is the clinical distinction between GR-active vs. GR-inactive lung cancer patients in terms of treatment responsiveness and long-term outcome?

We agree with the reviewer that it would have been interesting to provide *in vivo* analysis of the GR interactome, however this would involve a great deal of work and optimization since this has never been done on lung cancer tissue samples before, and ultimately clinical samples that we were unable to attain during the short revision time and COVID-19-related challenges.

To give more biomedical relevance we have now performed a Kaplan-Meier survival analysis of a lung cancer patient cohort consisting of 1529 cancer patients, of whom the majority did not receive (neo)adjuvant therapy before and after surgery. In this setting, patients with a high GR activity score had a more favorable overall survival and recurrence-free survival probabilities than patients with intermediate or low levels of GR activity. This can be appreciated in **Figure S1j-m**.

5. And: what about epidemiologic data on solid cancer incidence collected from long-term steroid-exposed patients, e.g. with autoimmune diseases or multiple myeloma?

This is indeed a very interesting question and we now highlighted this in the discussion section. Especially in context of lung cancer other researchers have performed large population-based studies and found that long-term steroid use, in particular inhaled corticosteroid use, may reduce risk of lung cancer development. The following citations were added:

Raymakers, Adam JN, et al. "Inhaled corticosteroids and the risk of lung cancer in COPD: a population-based cohort study." *European Respiratory Journal* 53.6 (2019).

Lee, Yu Min, et al. "Inhaled corticosteroids in COPD and the risk of lung cancer." *International journal of cancer* 143.9 (2018): 2311-2318.

Liu, Shih-Feng, et al. "Inhaled corticosteroids have a protective effect against lung cancer in female patients with chronic obstructive pulmonary disease: a nationwide population-based cohort study." *Oncotarget* 8.18 (2017): 29711.

Lee, Chang-Hoon, et al. "Inhaled corticosteroid use and risks of lung cancer and laryngeal cancer." *Respiratory medicine* 107.8 (2013): 1222-1233.

Kiri, Victor A., et al. "Inhaled corticosteroids and risk of lung cancer among COPD patients who quit smoking." *Respiratory medicine* 103.1 (2009): 85-90.

Parimon, Tanyalak, et al. "Inhaled corticosteroids and risk of lung cancer among patients with chronic obstructive pulmonary disease." *American journal of respiratory and critical care medicine* 175.7 (2007): 712-719.

Wang, I-Jen, et al. "Inhaled corticosteroids may prevent lung cancer in asthma patients." *Annals of thoracic medicine* 13.3 (2018): 156.

Minor concerns

1. Definitive wording such as "Here, we explain the molecular mechanisms of glucocorticoid response in solid cancer models" in the abstract is a heavy overstatement and should be omitted. "Here, we elucidate molecular..." would sound more appropriate.

We thank the reviewer for raising this issue. Following the reviewer's recommendation, we have reformulated the abstract now accordingly.

2. Experiments – in legends and the methods section – lack mandatory information; e.g. in Fig. 1b, c "cells treated with glucocorticoids (GC)" is simply insufficient: what steroid was used? At what dose? For how long?

We apologize for this omission. In the revised version of the manuscript, we now provide an additional supplementary table (Table S6), containing all the required information on steroid used, concentration and duration, for each panel. This approach was chosen, to limit space in the legends, but at the same time provide all the required data in a comprehensive fashion.

3. Fig. 1c "no signs of apoptosis" – where is the data?

We thank the reviewer for highlighting this omission. The statement was based on visual inspection of the time-course microscopy movies, but we agree that this was insufficient. Now, we provide a PARP western blot images for untreated and glucocorticoid treated cells. As you can appreciate in **Figure S1c**, GR activation did not induce PARP cleavage, indicating that apoptosis was not induced by GR activation. In addition, we have used IHC to monitor cleaved Caspase3 signal in the tumor samples of xenograft H1944 cells, which we were

unable to detect, with or without GR activation (**Figure S3i**). Furthermore, we have performed a FACS-based apoptotic assay in mesothelioma models, and also were unable to detect apoptosis in response to GR activation (**Figure S2c**).

4. Fig. 1d: since H1975 and H460 did not respond to GC with a proliferative slow-down, was there also no marked effect in terms of protein dephosphorylation upon GC? And the obvious question whether all five cell lines exhibit comparable GC/GR translocation to the nucleus is only addressed in Fig. 5a.

To address this comment, we have now added phospho-proteomic mass spectrometry analysis of these two cell lines. These figures representing these datasets can be found in **Figure S1d**. As it can be seen in the figure, glucocorticoids were unable to induce robust changes on phospho-proteome at standard difference and statistical cut-offs. This is in line with our findings that glucocorticoids do not induce growth arrest, and do not induce significant degree of gene expression modulation in these two cell systems as seen in **Figure 5c**. As for the GR nuclear translocation we fully understand the comment, yet still include this in the **Figure 5a**, as we feel it gets the point across better. We hope the reviewer understand this.

5. The paper is quite difficult to read due to its nebulous information about actual experimental conditions. For instance, Fig. 3d, claimed in the main text to show interaction of p57KIP2 with certain cell-cycle regulators, presents R1-4 (abbreviations for “technical replicates 1-4?”, conducted in an unknown cell line) as “Z-scaled PSM scores” – which to at least partially understand forces the reader to jump to the materials section.

We apologize for this particular example as well as other unclarities, and have extensively rewritten the manuscript to better clarify all assays and experimental in more details throughout the new version of the manuscript. As for the interactions of p57 with the co-regulators – the figure in the first version of the manuscript represents the data of four independent biological replicates in H2122 cell line which is used throughout the manuscript. However, to improve clarity we have now included the statistical comparison of p57-IP and corresponding IgG-IP, that shows various cell cycle related proteins to be enriched in our bait-IPs (**Figure 3d**). In addition, this altered visualization better shows specificity and not the whole proteome was enriched in our analysis. The analysis was done on data produced on four independent, biological replicates in H2122 cell line (and the raw data is deposited to PRIDE depository). We have now extensively updated descriptions in the result section, and legends.

6. I'm not sure that Fig. 3j, k convincingly shows that “...changes in E2F target genes and other cell-cycle-related genes typically induced by GCs were entirely absent in the p57-KO model, indicating that p57 is the sole driver in the growth arrest phenotype”.

We thank the reviewer for this comment, we have now performed additional analysis to address this remark. Firstly, we have included the statistics to substantiate the conclusions statistically (**Figure 3i-k**). Secondly, we have performed additional analysis using DESeq2, and now include a volcano plot, heatmap, and a gene-list analysis (*Raudvere, Uku, et al. "g: Profiler: a web server for functional enrichment analysis and conversions of gene lists (2019 update)." Nucleic acids research 47.W1 (2019): W191-W198.*) of all the genes changed between p57-WT and p57-KO in GC treatment condition (**Figure S5g-i**). The p57-KO cells treated with glucocorticoids do not exhibit statistically significant reduction in expression of various genes having a cell cycle-related role, as can be appreciated in **Figure 5Sg-i**, thus confirming the findings in the main figure. We have also edited the text, toning down our conclusions:

“While CRISPR-Cas9-mediated disruption of p57 did not alter transcriptional modulation of active-GR-associated genes (Fig. 3i, and Fig. S5g), it prevented downregulation of various genes involved in cell cycle (Fig. 5Sg-i). In conjunction with this, the changes in E2F target genes (Fig. 3j) and other cell cycle-related genes (Fig. 3k) typically induced by GCs were not observed in the p57-KO model, indicating that p57 upregulation is necessary for the growth arrest phenotype.

Reviewer #3 (Remarks to the Author):

Review Prekovic et al., Nature Communications 2020: “Glucocorticoid receptor triggers a novel mode of cell dormancy in solid cancers”.

The manuscript describes how glucocorticoids that are used to treat the side-effects of chemotherapy, might in fact also influence the behavior of the cancer cells by inducing a dormancy-like state. Using a chemical-screening approach, they show that this state makes cancer cells more susceptible to IGF-pathway inhibitors. Accordingly, in a xenograft model, co-treatment with GCs and an IGF inhibitor showed synergistic inhibition of tumor growth. Analysis of RNA-seq data from several cell lines in which GR induces this dormancy-like state unveils CDKN1C as candidate gene responsible for this phenotype. Several omic approaches show that this gene is indeed regulated by GR, requires interaction with SWI-SNF components and is likely regulated by a specific enhancer that “loops” to the CDKN1C promoter in response to GC treatment. Using genome-editing, the authors further show that induction of the dormancy-like state in cell lines requires an intact CDKN1C gene. Finally, analysis of TCGA data, specifically data regarding the chromatin accessibility of the enhancer they identify, shows quite convincingly that openness of this enhancer and CDKN1C expression correlate.

Overall, I think this manuscript is quite interesting and provides convincing evidence that the GC-dependent regulation of CDKN1C is responsible for the GC-induced dormancy-like state. It uncovers interesting details regarding the regulation of this gene by GR and about co-treatment with GCs and IGF-inhibitors as a potential treatment for solid cancers. With some modifications I think it would be publishable.

We would like to thank the reviewer for the valuable and constructive suggestions, and we are delighted to see the reviewer finds our work interesting and providing convincing evidence for the mechanism we studied. As can be appreciated below, all issues that were raised are addressed point-by-point, which clearly helped us to further improve the quality and clarity of our work.

Some remarks/Questions:

- What was the rationale for looking at the 5 cell lines you chose initially?

We thank the reviewer for this comment, as indeed the rationale for selecting these particular cell lines was not communicated clearly enough. We have selected these cell lines based on their steroid-hormone receptor expression profiles found in the literature. We now include a sentence addressing and the corresponding citations in the manuscript:

“In order to study the phenotypic and genotypic consequences of GR activation, five non-small cell lung cancer models were selected based on their steroid hormone receptor expression profiles^{14,15} (Fig. S1a).“

- Fig. 2a: I find this cartoon confusing. I think you either pre-treat or co-treat right? I am not sure if this cartoon explains the experimental set-up effectively....

We would like to thank the reviewer for highlighting this unclarity. We have now removed this panel, and added a new, more detailed one to the supplementary (**Figure S3a**) that also has a detailed legend. In addition, we have amended the results section including a better explanation on how the screen was performed.

- Related to this, how were synergistic effects defined in your study?

Excellent point. We have now changed the phrasing of ‘synergism’ into “enhancement”, as we do not provide the right methodology to justifiably claim synergism. We thank the reviewer for pointing this out.

- I don't think you have shown that p57 is the sole driver of the growth arrest phenotype as claimed on page 8. To make this claim, I think you would need to show that simply overexpressing p57 is enough to induce this state. I think you have shown that it is necessary, but not if it is sufficient.

We fully agree with the fact that we show that p57 is necessary for growth arrest induction (as shown in **Figure 3h**), but have not provided data that would suggest that p57 is sufficient for this arrest. To address this, we have now rephrased the text on page 8 to read:

"In conjunction with this the changes in E2F target genes (Fig. 3j) and other cell cycle related genes (Fig. 3k) typically induced by GCs were entirely absent in the p57-KO model, indicating that p57 upregulation is **necessary for** the growth arrest phenotype."

- Similarly, I do not agree that you have proven that the CERES enhancer is responsible for the regulation of the CDKN1C gene. I think your data (especially the TCGA data) provides very convincing indications that this is the case, but ultimate proof would require removing this enhancer to check if GR-dependent regulation still occurs in the absence of the CERES enhancer.

We sincerely appreciate this comment, and we fully agree with the reviewer that this experiment should be provided, in order to justifiably make this claim. To address the relevance of the reported enhancers on p57 regulation by glucocorticoids, we have now performed CRISPR-based enhancer perturbations to further investigate the mode of regulation of *CDKN1C*. Specifically, CRISPR-Cas9 was used to cut each of the three potential enhancer sites out of the genome, in which cutting of *CDKN1C* gene and *ABC1* promoter were used as positive and negative controls, respectively. The new data now shows that physical removal of enhancer 1 (CERES enhancer) significantly decreased GR-driven induction of *CDKN1C* expression in a polyclonal cell population, which is accompanied by a rescue of the cells from glucocorticoid receptor induced growth arrest. Importantly, we show that cutting of enhancer 2 and 3 do not lead to altered induction of growth arrest and *CDKN1C* induction. These findings can be appreciated in **Figures 4e and f**.

- Given that SWI-SNF components are cofactors for many TFs, it would be good to know if and how the knockdowns influence the basal expression level of the *CDKN1C* gene and not just the effect on the GC-induced level.

We thank the reviewer for highlighting this point. To address this, we have performed qPCR analysis of cells in the absence of glucocorticoids, to detect basal expression levels of *CDKN1C*. As it can be appreciated below, knocking down the SWI/SNF components had no effect on basal levels of *CDKN1C* expression.

Figure 2. Relative mRNA expression of *CDKN1C* after knock-down of SWI/SNF members in Vehicle-treated condition as determined by qPCR. No significant differences were found when compared to the control cell line.

Dunn's multiple comparisons test	Mean rank diff.	Significant?	Summary	Adjusted P Value	
shControl vs. shSMARCC2	3.063	No	ns	>0.9999	A-B
shControl vs. shSMARCB1	-3.688	No	ns	>0.9999	A-C
shControl vs. shSMARCA2	23.06	No	ns	0.1674	A-D
shControl vs. shSMARCD2	9.313	No	ns	>0.9999	A-E
shControl vs. shSMARCD3	21.31	No	ns	0.0812	A-F
shControl vs. shARID2	2.479	No	ns	>0.9999	A-G
shControl vs. shSMARCE1	15.15	No	ns	>0.9999	A-H
shControl vs. shARID1A	3.813	No	ns	>0.9999	A-I

- The results of the TCGA analysis showing that the openness of the CERES enhancer correlates very strongly with expression of CDKN1C is really nice. Any SNPs linked to cancer susceptibility mapping to this locus?

We are happy to see the reviewer appreciates this analysis. This is indeed an interesting question. Unfortunately, non-of the five SNPs mapped to our enhancer (rs2237884, rs233434, rs5789271, rs12794000, and rs234886) seem to have a link to cancer susceptibility. In addition to this, we have looked into various GWAS and out of these SNPs the rs234886 has the most compelling link with human height (figure below). Interestingly, a recent study also found height-related SNPs (including a SNP in perfect LD with rs234886) to be associated with lung cancer risk in east Asian population (Wang, Lu, et al. "Genetically determined height was associated with lung cancer risk in East Asian population." *Cancer medicine* 7.7 (2018): 3445-3452.). This is definitely of interest, but we do feel this lies beyond the scope of the current work.

Figure 3.

A ranked plot showing association of rs234886 across different GWAS.

- Maybe you could add some speculation to the discussion regarding the nature etc of this novel mode of dormancy you propose. How is it different from other types of dormancy and how might this be relevant?

We have now amended the discussion and further discuss the components of the phenotype reported and their potential role. However, following the suggestions of Reviewer 1 we now describe the cellular state induced as a “reversible drug-tolerant dormancy state”, not as “hybernescence”.

Some typos:

- Line 65: Incorrect reference to figure?! I think this should be Figure S1C

- Line 222: referred should be refered.

- Line 326: Remove “been”

We have edited the manuscript with care to ensure no typos and grammatical errors are present. Thank the reviewer for highlighting these typos, which have now all been corrected.

REVIEWERS' COMMENTS

Reviewer #1 (Remarks to the Author):

Comment on Zwart et al.

The authors did a good job addressing my concerns, most importantly moderating the language and claims to better match the data. My remaining major concern is that I do not see the authors giving credit to Suzanne Conzen's work which also recommends caution when administering glucocorticoids as an anti-emetic following solid-tumor chemotherapy. This is mentioned in the abstract, but not included in the intro or discussion. This is important as if glucocorticoids are used in combination with other chemotherapeutics they may blunt their effect even though they might have some beneficial effect on the tumors when administered alone.

There are a few quibbles I have and are listed below, but they are relatively minor.

Line 63: Glucocorticoids are used to treat *lymphoid* cancer - not hematological broadly (e.g. not AML)

Line 71: To say a "precise" mode of action is "entirely" unknown is odd. Perhaps "not clear" instead?

Line 109: "most of which were involved in direct regulation of the cell cycle" - From Figure 1e it appears "most" affect transcription.

Line 132: Question - why are the affected cells and unaffected cell phosphoproteome differences presented differently? Only proteins that decrease in phosphorylation are displayed for the affected cells - why not do a volcano plot for these like S1d? Particularly in light of GC-induced phosphorylation of some important factors (IGF-1R - line 271) this would be useful to present.

Line 134 - "no signs of apoptosis were detected" implies that more than one sign (cleaved parp) was probed for. It is more accurate to say "The lack of cleaved PARP suggests that growth arrest does not involve apoptosis"...or something like that.

It is difficult to come away with any conclusion from Figure S3b as it is laid out.

Line 364 - there is no difference between "critically required" and "required". A gene is either required or it is not.

Line 408 - the enhancer is not "novel"...it may have been previously uncharacterized, but it was always there.

Suggestion - Line 445 - rather than "we inspected the direct regulation of the CDKN1C gene by GR using ChIP sequencing" which sounds like direct regulation has already been established, maybe "we sought to establish direct regulation by GR and to elaborate on the mechanism using ChIP sequencing"

Suggestion - Line 462 - 4C doesn't quite "determine whether these particular enhancers and the CDKN1C locus directly interact in 3D genome space" it establishes proximity. Direct interaction would show that they touch - which is very hard to prove. I would suggest avoiding saying that there is direct interaction or physical contact and instead say that they are in the proximity or enhanced proximity.

Lovely experiments using CRISPR to probe enhancers!!!

Line 701: "However, direct causality, linking particular genes to phenotypes, is still understudied." It would be nice to give some credit to folks who have done this well here, e.g. Kruth et al., Blood 2017.

Line 736: "Our comprehensive GR transcriptional complex" - is this referring to RIME + expression? I would not call this comprehensive. As you say above, a functional characterization of all interacting proteins and genes is needed, but not done here. This language should be softened.

I might also trumpet the dissection of the roles for enhancers in CDKN1C regulation by GR. There are too few examples of this being done carefully in the literature - people need to see this as an example.

Nice work - Miles Pufall

Reviewer #2 (Remarks to the Author):

Prekovic-S,... ..Zwart-W, Glucocorticoid receptor triggers a novel mode of cell dormancy in solid cancers
Revised version submitted to Nature Communications

This is now the revised version of the manuscript by Zwart and colleagues on the anti-proliferative glucocorticoid (GC) effects mediated via IGF-1R signaling and the CDKN1C locus (encoding p57KIP2), specifically through binding of a glucocorticoid receptor (GR)/histone acetyltransferase p300 complex to an enhancer element (named CERES – CDKN1C Enhancer Regulated by Steroids) in five non-small cell lung cancer (NSCLC) cell lines. The paper specifically focuses on p57KIP2-mediated GC action via direct genomic interaction of the GR at the CDKN1C locus; other GC effects on cell-cycle regulation remained and remain largely unaddressed. However, the authors convincingly demonstrate the biological relevance of their focused investigation.

The substantial revision of this bold manuscript indeed improved the quality of the work and its central scientific messages significantly. Especially the efforts undertaken to selectively eliminate candidate CDKN1C regulatory elements by CRISPR/Cas9 technique (i.e. the CERES element) are mechanistically important and insightful.

Moreover, rephrasing of text sections and legends balanced overstatements and helped to enhance clarity (which was a major issue of the initial version). Nevertheless, editorial support is still needed to present this work in a scientifically sound and understandable way.

In essence, I'm satisfied with the revised version and would recommend publication in Nature Communications.

Minor

1. The authors now toned down on FoxO1 and IRS2 mediating GR impact on the IGF-1R pathway – by moving the data to the supplements. The functional investigation of this claim remains weak; the reviewer-only data (Review-only Fig. 1), stating that inhibition of FoxO1 or IRS2 would diminish GC effects in the IGF-1R cascade, is not convincing.

Reviewer #3 (Remarks to the Author):

Review revised manuscript Prekovic et al., : "Glucocorticoid receptor triggers a reversible drug-tolerant dormancy state with acquired therapeutic vulnerabilities in lung cancer".

Each of my initial comments was addressed to my satisfaction. In addition, the study was expanded e.g. by adding more insights regarding the cis regulatory elements involved in regulation of the CDKN1C gene which adds to its appeal

Some last small suggestions/remarks:

Abstract: In the abstract the authors caution for use of glucocorticoids in the treatment of side-effects of anti-cancer treatment. However, the message I take away for this study is rather that GR activity (and even GC treatment) is overall likely to be beneficial despite the fact that it induces a dormancy state that makes cells less sensitive to various anti-cancer drugs. Moreover, the authors provide a possible drug that might target the cells in this dormancy state which could eventually improve therapies. My interpretation could of course be wrong, but if not, I think the findings of the paper would be captured better when the abstract would be revised to capture the complexity of the use of GC in treatment.

Abstract: line 23: I think "underlines" should be "underlies"

Line 75: response TO a large array of anti-cancer drugs (to missing in sentence)

Lack of cleaved PARP (Fig. S1c): Here adding a positive control, e.g. using a drug to induce apoptosis and PARP cleavage, would make the results more convincing.

Line 186: Reversible state: Question: is this reversible statement based on the fact that GC do not induce apoptosis? If so state this explicitly, as I think the authors have not shown that removal of GCs results in reversing the dormancy state.

Pages 13-15: Link Swi/Snf to GR-dependent gene regulation in mammalian cells has been described in several other studies, would be good to include some references to acknowledge this (e.g. PMID 18342607). Along the same line, distinct swi/snf complexes with specific functions have been described (e.g. search for papers by Gerald Crabtree) and it would be good to include this info, e.g. on page 20 in the discussion....

Line 686: amd should be and

Reviewer #1 (Remarks to the Author):

Comment on Zwart et al.

The authors did a good job addressing my concerns, most importantly moderating the language and claims to better match the data. My remaining major concern is that I do not see the authors giving credit to Suzanne Conzen's work which also recommends caution when administering glucocorticoids as an anti-emetic following solid-tumor chemotherapy. This is mentioned in the abstract, but not included in the intro or discussion. This is important as if glucocorticoids are used in combination with other chemotherapeutics they may blunt their effect even though they might have some beneficial effect on the tumors when administered alone.

We would like to thank the reviewer for appreciating the efforts we put in. To address this comment, we have added additional citations from the Conzen group:

"While decrease in sensitivity to selected chemotherapeutics after GC treatment has been observed previously^{43,67-70}, we have unbiasedly profiled a large number of compounds to show that this generally applies to various drugs."

There are a few quibbles I have and are listed below, but they are relatively minor.

Line 63: Glucocorticoids are used to treat *lymphoid* cancer - not hematological broadly (e.g. not AML)

This has been corrected. Word "hematological" has been replaced with lymphoid.

Line 71: To say a "precise" mode of action is "entirely" unknown is odd. Perhaps "not clear" instead?

This has been corrected:

"Despite these observations, a precise mode-of-action through which GCs affect non-lymphoid solid cancers remains unclear."

Line 109: "most of which were involved in direct regulation of the cell cycle" - From Figure 1e it appears "most" affect transcription.

We have changed the text accordingly:

"most of which were involved in direct regulation of transcription and cell cycle as evidenced by geneset analysis"

Line 132: Question - why are the affected cells and unaffected cell phosphoproteome differences presented differently? Only proteins that decrease in phosphorylation are displayed for the affected cells - why not do a volcano plot for these like S1d? Particularly in light of GC-induced phosphorylation of some important factors (IGF-1R - line 271) this would be useful to present.

We did not want to focus on the sites have increase in phosphorylation in main Figure 1, as that comes later in the manuscript (Figure 2), therefore we kept it as it was.

Line 134 - "no signs of apoptosis were detected" implies that more than one sign (cleaved parp) was probed for. It is more accurate to say "The lack of cleaved PARP suggests that growth arrest does not involve apoptosis"...or something like that.

This has been amended:

"As lack of cleaved PARP suggested that growth arrest does not involve apoptosis (Supplementary Fig. 1c), we inspected whether GC-treatment led to acquisition of senescence."

It is difficult to come away with any conclusion from Figure S3b as it is laid out.

We have made changes to Supplementary Fig. 3b in order to help with the interpretation.

Line 364 - there is no difference between “critically required” and “required”. A gene is either required or it is not.

The word “critically” has been removed.

Line 408 - the enhancer is not “novel”...it may have been previously uncharacterized, but it was always there.

The word novel has been removed, and replaced with “previously uncharacterized”.

Suggestion - Line 445 - rather than “we inspected the direct regulation of the CDKN1C gene by GR using ChIP sequencing” which sounds like direct regulation has already been established, maybe “we sought to establish direct regulation by GR and to elaborate on the mechanism using ChIP sequencing”

This has been corrected accordingly.

“Therefore, we sought to establish direct regulation of the CDKN1C gene by GR and to elaborate on the mechanism using ChIP sequencing.”

Suggestion - Line 462 - 4C doesn't quite “determine whether these particular enhancers and the CDKN1C locus directly interact in 3D genome space” it establishes proximity. Direct interaction would show that they touch - which is very hard to prove. I would suggest avoiding saying that there is direct interaction or physical contact and instead say that they are in the proximity or enhanced proximity.

Lovely experiments using CRISPR to probe enhancers!!!

This has now been implemented:

“To establish whether these particular enhancers and the CDKN1C locus are in proximity to one another in 3D genome space, we performed 4C-seq experiments³⁷. The unbiased interaction analyses from the viewpoint of the CDKN1C promoter, revealed that GC treatment enhanced the interaction with two distal regions within the KCNQ1 gene (a and b) (Fig. 4c). Region a and b coincided with locations of CERES and enhancer 2/3, respectively. To univocally show that signal originating from region a is driven by close proximity of CDKN1C promoter and CERES, we performed the reciprocal 4C-seq experiment from the CERES viewpoint. In A549, H2122, and H1944, we observed a statistically significant enhancement of the contact between this enhancer and CDKN1C promoter by GCs (Fig. 4d). Conversely, this enhancement was absent in two GC-unresponsive models of lung cancer, H1975 and H460 cell lines (Supplementary Fig. 6b).”

Line 701: “However, direct causality, linking particular genes to phenotypes, is still understudied.” It would be nice to give some credit to folks who have done this well here, e.g. Kruth et al., Blood 2017.

This has now been corrected:

“However, direct causality, linking particular genes to phenotypes, is still understudied, with only a few examples in the literature.”

And the appropriate literature has been cited.

Yang, H. et al. Stress–glucocorticoid–TSC22D3 axis compromises therapy-induced antitumor immunity. *Nat. Med.* 25, 1428–1441 (2019).

Abraham, S. M. et al. Antiinflammatory effects of dexamethasone are partly dependent on induction of dual specificity phosphatase 1. *J. Exp. Med.* 203, 1883–1889 (2006).

Kruth, K. A. et al. Suppression of B-cell development genes is key to glucocorticoid efficacy in treatment of acute lymphoblastic leukemia. *Blood, J. Am. Soc. Hematol.* 129, 3000–3008 (2017).

Line 736: “Our comprehensive GR transcriptional complex” - is this referring to RIME + expression? I would not call this comprehensive. As you say above, a functional characterization of all interacting proteins and genes is needed, but not done here. This language should be softened.

We have removed the word “comprehensive” to soften the statement.

I might also trumpet the dissection of the roles for enhancers in CDKN1C regulation by GR. There are too few examples of this being done carefully in the literature - people need to see this as an example.

We have now included a sentence in the discussion section:

“Importantly, we functionally probed this enhancer using CRISPR-Cas9-based deletion to demonstrate that this enhancer is critically involved in GR-driven CDKN1C expression.”

Nice work - Miles Pufall

Reviewer #2 (Remarks to the Author):

Prekovic-S,... ...Zwart-W, Glucocorticoid receptor triggers a novel mode of cell dormancy in solid cancers
Revised version submitted to Nature Communications

This is now the revised version of the manuscript by Zwart and colleagues on the anti-proliferative glucocorticoid (GC) effects mediated via IGF-1R signaling and the CDKN1C locus (encoding p57KIP2), specifically through binding of a glucocorticoid receptor (GR)/histone acetyltransferase p300 complex to an enhancer element (named CERES – CDKN1C Enhancer Regulated by Steroids) in five non-small cell lung cancer (NSCLC) cell lines. The paper specifically focuses on p57KIP2-mediated GC action via direct genomic interaction of the GR at the CDKN1C locus; other GC effects on cell-cycle regulation remained and remain largely unaddressed. However, the authors convincingly demonstrate the biological relevance of their focused investigation.

The substantial revision of this bold manuscript indeed improved the quality of the work and its central scientific messages significantly. Especially the efforts undertaken to selectively eliminate candidate CDKN1C regulatory elements by CRISPR/Cas9 technique (i.e. the CERES element) are mechanistically important and insightful.

Moreover, rephrasing of text sections and legends balanced overstatements and helped to enhance clarity (which was a major issue of the initial version). Nevertheless, editorial support is still needed to present this work in a scientifically sound and understandable way.

In essence, I'm satisfied with the revised version and would recommend publication in Nature Communications.

Minor

1. The authors now toned down on FoxO1 and IRS2 mediating GR impact on the IGF-1R pathway – by moving the data to the supplements. The functional investigation of this claim remains weak; the reviewer-only data (Review-only Fig. 1), stating that inhibition of FoxO1 or IRS2 would diminish GC effects in the IGF-1R cascade, is not convincing.

We agree with the reviewer and future work should indeed go deeper into understanding how GCs alter the IGF-1R signaling.

Reviewer #3 (Remarks to the Author):

Review revised manuscript Prekovic et al., :“Glucocorticoid receptor triggers a reversible drug-tolerant dormancy state with acquired therapeutic vulnerabilities in lung cancer”.

Each of my initial comments was addressed to my satisfaction. In addition, the study was expanded e.g. by adding more insights regarding the cis regulatory elements involved in regulation of the CDKN1C gene which adds to its appeal

We thank the reviewer for the positive evaluation, and we are happy that the reviewer was satisfied with our response.

Some last small suggestions/remarks:

Abstract: In the abstract the authors caution for use of glucocorticoids in the treatment of side-effects of anti-cancer treatment. However, the message I take away for this study is rather that GR activity (and even GC treatment) is overall likely to be beneficial despite the fact that it induces a dormancy state that makes cells less sensitive to various anti-cancer drugs. Moreover, the authors provide a possible drug that might target the cells in this dormancy state which could eventually improve therapies. My interpretation could of course be wrong, but if not, I think the findings of the paper would be captured better when the abstract would be revised to capture the complexity of the use of GC in treatment.

The reviewer raises an interesting, yet complicated question, as the ‘caution for the use of glucocorticoids’ statement was included in the abstract, to address a concern about this issue from Reviewer #1. However, we do understand the point this reviewer highlights, and reformulated the abstract accordingly, now better highlighting the complexity of GR activation in this context, both raising caution when using standard therapeutics but at the same time presenting opportunities by induction of acquired vulnerabilities.

Abstract: line 23: I think “underlines” should be “underlies”

This has been corrected.

Line 75: response TO a large array of anti-cancer drugs (to missing in sentence)

This has been corrected.

Lack of cleaved PARP (Fig. S1c): Here adding a positive control, e.g. using a drug to induce apoptosis and PARP cleavage, would make the results more convincing.

In retrospect, this is indeed something that should have been done. We have used a validated (cleaved) PARP antibody, used in various publications. In addition to that we have also used an Annexin-based apoptosis assay in which we have included a positive control, and observing absence of signal for the glucocorticoid-treated condition.

Line 186: Reversible state: Question: is this reversible statement based on the fact that GC do not induce apoptosis? If so state this explicitly, as I think the authors have not shown that removal of GCs results in reversing the dormancy state.

In the revised version of the manuscript we have included live-cell imaging (confluency) measurements of proliferation through which we show that cells re-start proliferation once the ligand is removed (Supplementary Fig. 1f).

Pages 13-15: Link Swi/Snf to GR-dependent gene regulation in mammalian cells has been described in several other studies, would be good to include some references to acknowledge this (e.g. PMID 18342607). Along the same line, distinct swi/snf complexes with specific functions have been described (e.g. search for papers by Gerald Crabtree) and it would be good to include this info, e.g. on page 20 in the discussion....

We have now included additional citations.

Biggar, S. R. & Crabtree, G. R. Continuous and widespread roles for the Swi-Snf complex in transcription. *EMBO J.* 18, 2254–2264 (1999).

Bultman, S. et al. A Brg1 null mutation in the mouse reveals functional differences among mammalian SWI/SNF complexes. *Mol. Cell* 6, 1287–1295 (2000).

Wang, W. et al. Diversity and specialization of mammalian SWI/SNF complexes. *Genes Dev.* 10, 2117–2130 (1996).

Hoffman, J. A., Trotter, K. W., Ward, J. M. & Archer, T. K. BRG1 governs glucocorticoid receptor interactions with chromatin and pioneer factors across the genome. *Elife* 7, e35073 (2018)

Line 686: amd should be and

This has been corrected.